# Causal networks for climate model evaluation and constrained projections

Peer Nowack [1,2,3,4✉], Jakob Runge [5,1], Veronika Eyring [6,7] & Joanna D. Haigh [1,2]

Global climate models are central tools for understanding past and future climate change. The assessment of model skill, in turn, can benefit from modern data science approaches. Here we apply causal discovery algorithms to sea level pressure data from a large set of climate model simulations and, as a proxy for observations, meteorological reanalyses. We demonstrate how the resulting causal networks (fingerprints) offer an objective pathway for process-oriented model evaluation. Models with fingerprints closer to observations better reproduce important precipitation patterns over highly populated areas such as the Indian subcontinent, Africa, East Asia, Europe and North America. We further identify expected model interdependencies due to shared development backgrounds. Finally, our network metrics provide stronger relationships for constraining precipitation projections under climate change as compared to traditional evaluation metrics for storm tracks or precipitation itself. Such emergent relationships highlight the potential of causal networks to constrain long-standing uncertainties in climate change projections.

[1] Grantham Institute, Imperial College London, London SW7 2AZ, UK. [2] Department of Physics, Faculty of Natural Sciences, Imperial College London, London SW7 2AZ, UK. [3] Data Science Institute, Imperial College London, London SW7 2AZ, UK. [4] School of Environmental Sciences, University of East Anglia, Norwich NR4 7TJ, UK. [5] German Aerospace Center, Institute of Data Science, 07745 Jena, Germany. [6] Deutsches Zentrum für Luft- und Raumfahrt (DLR), Institut für Physik der Atmosphäre, Oberpfaffenhofen, Muenchener Strasse 20, 82234 Wessling, Germany. [7] University of Bremen, Institute of Environmental Physics, 28359 Bremen, Germany. ✉email: p.nowack@uea.ac.uk

State-of-the-art climate and Earth system models represent an enormous scientific achievement and are central tools to understand past climates, as well as to project future climate change. More than 40 modelling centres worldwide undertake climate model development[1–3], and have rapidly elevated their level of sophistication. Nowadays, many models simulate not only fundamental physical laws of fluid motion, energy and momentum conservation, but also include interactive carbon cycle, aerosol, and atmospheric chemistry schemes, or resolve the entire stratosphere[4–10]. However, while all climate models are based on the same physical principles, there are development-specific choices that lead to significant model differences, in particular related to subgrid-scale parameterisations of clouds, convection and aerosols[11–13]. These contribute to persistent discrepancies between models and observations, as well as among model projections, for example, regarding precipitation changes[1,14,15]. Multi-model evaluation and intercomparison is often based on the mean and variance of aggregate quantities, such as temperature, or spectral properties and (auto-)correlation measures[16–18]. One issue with such metrics is that models can be right for the wrong reasons due to offsetting biases[11,12,16].

Here, we introduce causal model evaluation (CME) as a type of process-oriented model evaluation[11,18–20]. CME deploys recently developed causal discovery methods[21–23] adapted for applications to climate data[23–27]. Within the CME framework, we evaluate the ability of models from the Coupled Model Intercomparison Project Phase 5 (CMIP5) to simulate atmospheric dynamical interactions classically measured as lagged correlations between climate variables at remote locations[28–31]. Causal discovery algorithms go beyond correlation-based measures by systematically excluding common driver effects and indirect links[22,26,32,33]. We show that characteristic causal fingerprints can be learned from climate data sets, which are robust among ensemble members of the same model and, for example, can identify shared model development backgrounds. Fingerprints closer to observations are also associated with smaller precipitation biases in climate models. Finally, we highlight the potential of our approach to offer a pathway to reducing uncertainties in climate change projections, as well as to understand differences between models and observations.

## Results

**Causal model evaluation framework**. To characterise the network of global dynamical interactions, we use a causal discovery algorithm to reconstruct directed, time-lagged interdependency networks from global climate data sets. Figure 1 provides an overview of the individual steps of the CME framework (see Methods for details).

The selection of components defining the network nodes will typically be guided by expert knowledge in conjunction with dimension reduction techniques. Here, we use components obtained through Varimax-rotated principal component analysis[34,35] (PCA) applied to sea level pressure anomaly data (Fig. 1a; Methods). For sea level pressure data, PCA-Varimax components can be interpreted as major modes of climate variability[25,28,36,37]. Owing to the seasonal character of interaction pathways[28,38], we construct individual components, and in the next step networks, for the four meteorological seasons: December, January, February (DJF); March, April, May (MAM); June, July, August (JJA); September, October, November (SON). We select 50 components for each season (Methods) whose geographic locations for DJF are indicated in Fig. 1b (for all seasons see Supplementary Fig. 1). PCA-Varimax can identify the major modes of variability[37], for example, related to the El Niño

Southern Oscillation (ENSO) in the East, West and Central Pacific[39] (components 1,4,5 in Fig. 1b).

We calculate interactions among these nodes as causal networks from the associated component time series (Fig. 1b). For this step, we use the PCMCI algorithm by Runge et al.[23,26], which is particularly suited for high-dimensional and auto-correlated climate data (Methods). In contrast to pure correlation measures, causal discovery methods are built to remove spurious links due to common drivers and indirect pathways from the networks (Fig. 1c)[22,26]. The resulting networks contain information on the direction and associated time lags of potential causal links, characterising the pathways of the global interaction network. PCMCI has been tested extensively to successfully recover important interactions in the climate system, such as the tropical Walker circulation and predictors of polar vortex states[23,24,26,27]. Note that, in these network structures, some established interactions measured traditionally as direct correlations between climate modes can follow a more complex pathway of indirect links. We illustrate this for the coupling between ENSO and the Pacific-South American (PSA) pattern[29,40] in Supplementary Fig. 2.

The resulting causal networks effectively represent characteristic causal fingerprints[41,42] for each sea level pressure data set (Fig. 1d), which can be compared using network metrics[25]. Each network consists of hundreds of links. Generally, we conduct pair-wise comparisons of all possible links in a network A to a network B, taking A as the reference network. For example, we test if a link from component 4 (West Pacific ENSO) to component 1 (East Pacific ENSO) found in observations is also detected in climate model data sets. We use a modified asymmetric $F_1$-score (Methods) as the harmonic mean of precision (fraction of links in B that also occur in A) and recall (fraction of links in A that are detected in B). $F_1$-scores vary between 0 and 1 (perfect network match). The network comparison results depend on the number of links considered to be statistically significant (Methods). However, we tested that all conclusions based on the 400–500 most significant links per network included here are robust to a large range of possible network link densities from a hundred to more than a thousand links (Supplementary Figs. 3–6 and Supplementary Table 1).

**Application to pre-industrial simulations**. Pre-industrial simulations are well suited for the CME of atmospheric dynamical interactions due to the many years simulated by each model in the absence of transient effects caused by anthropogenic forcings[1–3]. Specifically, we applied the CME framework to 210 years of global DJF sea level pressure data from each of in total 20 CMIP5 models at a 3-day time resolution (Methods; Fig. 2). In our algorithm settings, we include interactions on a time-scale of up to 30 days ($\tau_{max} = 10$; Methods). We split each 210-year data set into three 70-year intervals (ensemble members) to study multi-decadal variations[43,44]. As a result, we obtain nine possible network comparisons for each pair of models and six distinct comparisons between ensemble members of the same model. $F_1$-scores for these model intercomparisons are shown in Fig. 2. Three major features highlight the skill of the CME framework.

Firstly, each model can be recognised individually purely based on its causal fingerprint. Networks estimated from different ensemble members of the same model are more consistent than networks estimated from two different models as evident from the high $F_1$-scores on the diagonal of the matrix in Fig. 2a (dark red). Each row in Fig. 2a denotes the model used as the reference against which each column is compared.

Secondly, models with shared development background can be detected. Many climate models share software, resulting in

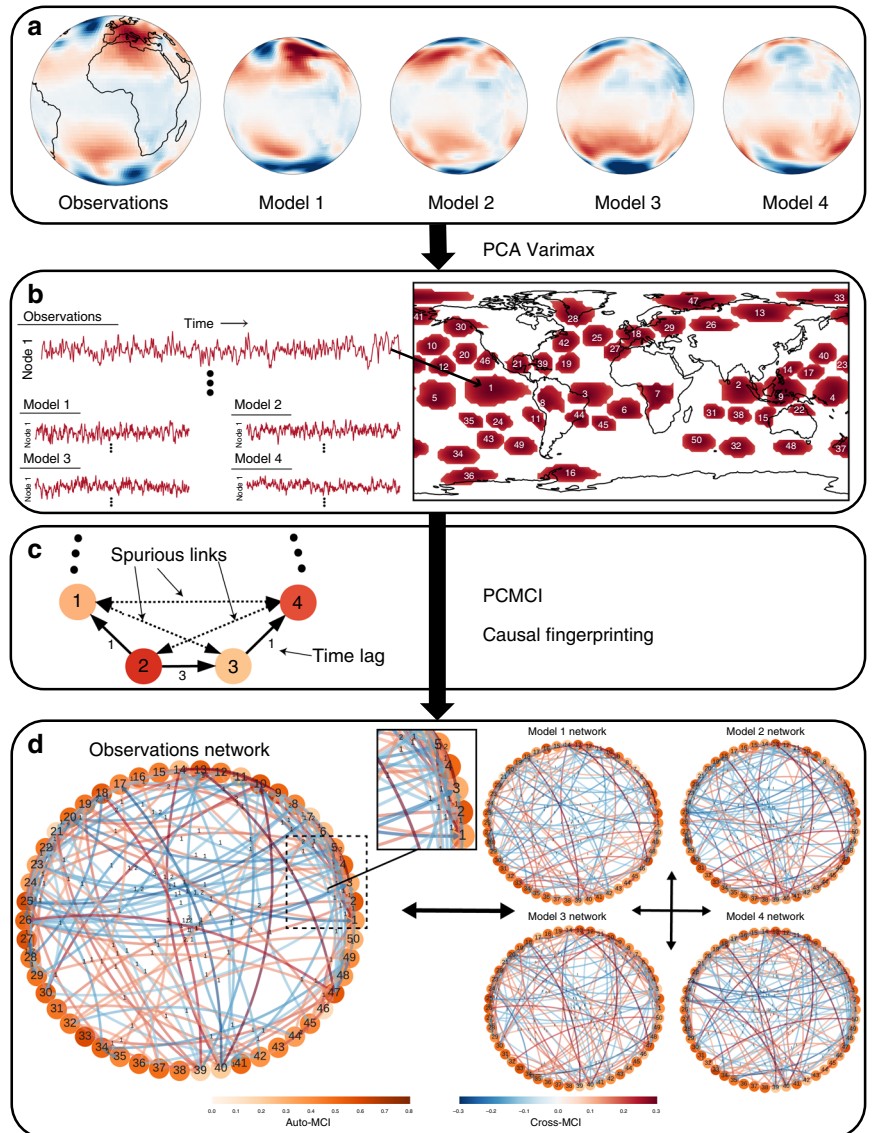

**Fig. 1 Sketch of the causal model evaluation framework. a** Gridded Earth system data, here daily-mean sea level pressure from the NCEP-NCAR reanalysis (approximating observations)[52], is dimension-reduced using PCA-Varimax to (**b**) a set of regionally confined climate modes of variability. The same transformation is subsequently applied to climate model data (Methods). Core component regions (in this case for the season December–January–February) are indicated in red. Each component is associated with a time series and serves as one of the network nodes. Here, the component time series are afterwards 3-day-averaged. **c** PCMCI estimates directed lagged links among these nodes giving rise to (**d**) data set-characteristic causal fingerprints, which can be used for model evaluation and intercomparison. Node colours in **d** indicate the level of autocorrelation (auto-MCI) as the self-links of each component and link colours the interdependency strength (cross-MCI). Link-associated time lags (unit = 3 days) are indicated by small labels. Only the around 200 most significant links each for the reanalysis and for data from 4 climate models are shown. Links with lag zero, for which directions cannot be easily causally resolved, are not shown.

important interdependencies among them[12,45–49]. CME can detect such shared backgrounds (highlighted by black squares in Fig. 2a). For example, CME identifies the models HadGEM2-ES, HadGEM2-CC, ACCESS1-0 and ACCESS1-3 as similar, which are all versions of the HadGEM model family[50,51] developed by the UK Met Office. There is a clear separation between these four and the remaining models, see Fig. 2b showing all scores when HadGEM2-ES networks are taken as the reference. The different models developed by the Institute Pierre Simon Laplace (IPSL), the Max-Planck Society (MPI) and the Geophysical Fluid Dynamics Laboratory (GFDL) are also each recognised as subgroups (Fig. 2c–e). For the Japanese MIROC models, two out of three are detected as a subgroup (MIROC-

ESM, MIROC-ESM-CHEM), whereas MIROC5 is even less similar than the multi-model average (grey line in Fig. 2f). We conclude that CME can detect similar models, a condition often but, as shown here, not always synonymous with models developed under the same research umbrella. This demonstrates the significant potential of using CME to assess model interdependencies based on causal networks.

Thirdly, climate models are recognised to share a physical ground truth. We further compared all 20 models with two artificial reference cases: Random and Independent (last two rows/columns in Fig. 2a; Methods). For Random, we created 50 randomly coupled and auto-correlated noise time series, i.e., there are links in the system, but these do not follow any Earth system

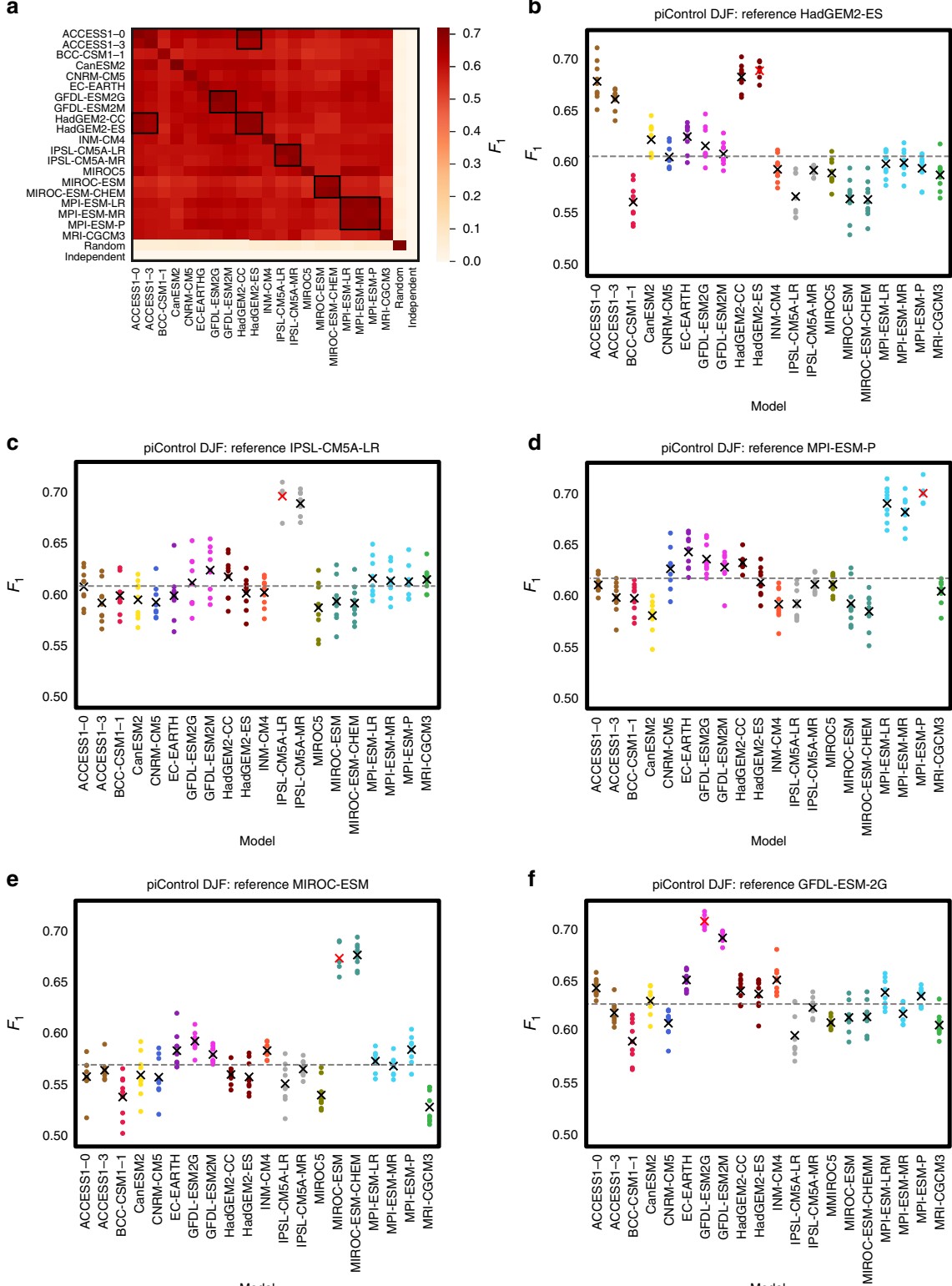

**Fig. 2 Pre-industrial network intercomparison scores. a** Matrix of average $F_1$-scores for pair-wise network comparisons between ensemble members of 20 climate models (labelled following CMIP5 nomenclature in capital letters) using data for December–January–February (DJF) and the two surrogate models (Random, Independent). Rows are models taken as reference in each case, columns are the models that are compared to these references. Higher scores imply better agreement between networks, i.e., that two models are more similar in terms of their causal fingerprint. **b–f** Scatter plots showing each individual network comparison score, with different models taken as reference (as labelled in the sub-figure titles) that the other models (labelled on the x-axis using capital letters) are compared to. Black crosses (red for the reference) mark average results also shown in (**a**). Grey dashed lines mark the average score excluding the reference itself. Our causal model evaluation approach detects the expected similarities between certain model groups as shown in (**b–f**), which are additionally indicated by inset black squares in (**a**). Source data are provided as a Source Data file.

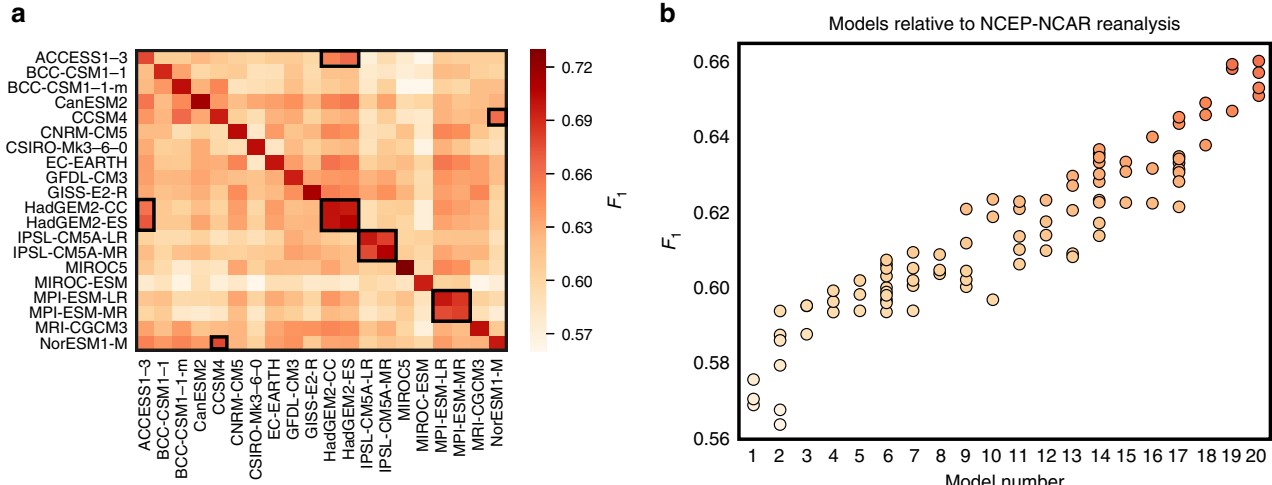

**Fig. 3 Historical network comparisons. a** As Fig. 2a, but for climate model simulations spanning approximately the historical period from 1st January 1948 to 31st December 2017 for which 20 CMIP5 models with up to ten different ensemble members are available. **b** Ordered $F_1$-scores when the causal fingerprint learned from NCEP-NCAR reanalysis data is taken as the reference. Differences in **b** are highly statistically significant, with $p$-values $< 9 \times 10^{-10}$ for a non-parametric Kruskal–Wallis-test and $p < 5 \times 10^{-30}$ for a standard one-way ANOVA $F$-test. The model key for **b** is provided in Supplementary Table 1. We note that similar model rankings have been found regionally for precipitation, e.g., for China[55]. Individual network scores (marker colours) in **b** follow the colour code from (**a**). Source data are provided as a Source Data file.

physics. As evident from Fig. 2a, the corresponding networks are self-consistent (diagonal entry) but achieve very low $F_1$-scores when compared to the actual climate models. For Independent, we created auto-correlated time series without any significant coupling among them so that any detected links occur randomly in the system (false positives). CME expectedly finds low scores throughout for this case.

**Causal model evaluation of historical simulations.** Motivated by CME's skill to recognise models with shared development background, we next evaluate the CMIP5 models with NCEP-NCAR reanalysis data[52] as a proxy for recent observations. We calculate fingerprints from 20 CMIP5 simulations covering approximately the historical period from 1st January 1948 to 31st December 2017 (Methods). For better statistical estimates, we only included models for which at least three ensemble members were available (Supplementary Table 2). To additionally investigate the role of seasonal variability, we carried out separate analyses for DJF, MAM, JJA and SON. However, all seasons yielded very similar results (Supplementary Figs. 3–6) and we focus the discussion on annual $F_1$-scores averaged over all seasons (Methods).

We find effectively the same model subgroups as before (inset boxes in Fig. 3a). Owing to the slightly different setup, there is an additional subgroup related to the climate model CCSM4 (Supplementary Fig. 7). Taking the NCEP-NCAR reanalysis network as the reference, we obtain an estimate of how well individual models capture the observed causal fingerprint (Fig. 3b; the models are ordered by average $F_1$-score). The result is a continuum rather than a clear-cut differentiation between a better and a worse group of models. However, models do exhibit significantly different causal fingerprints ($p$-value[53] $< 10^{-9}$). We conducted the same analysis using a shorter ERA-Interim reanalysis data set[54] to estimate the reference network and obtained almost the same model order (Supplementary Fig. 8 and Supplementary Table 1).

**Implications for precipitation modelling.** Atmospheric dynamical interactions as imprinted here on the sea level pressure field are well-known drivers of precipitation anomalies in many world

regions[28,29]. Therefore, we test for relationships between the reanalysis-referenced $F_1$-scores of CME and Taylor $S$-scores[55,56] for precipitation rates, which measure grid-cell-wise errors in conjunction with overall discrepancies in precipitation variability across a spatial domain. To calculate the $S$-scores, which also range from 0 to 1, we use historical Climatic Research Unit (CRU)[57] land surface precipitation data from the University of East Anglia, averaged over the years 1948–2017 (Methods).

We find that better fingerprints are associated with smaller land precipitation biases ($F_1$- and $S$-scores are positively correlated; Fig. 4a). This is true globally (correlation coefficient $R = 0.7$), as well as in many world regions known to be influenced by (remote) dynamical interactions, in particular North America ($R = 0.7$), East Asia ($R = 0.6$), Africa ($R = 0.5$) and South Asia ($R = 0.5$). These results also hold if we disregard models belonging to the same subgroups as marked in Fig. 3a. There are some regional exceptions (e.g., Australia, Indonesia) where we find no significant correlations. A possible explanation is predominant regional factors[17,39] rendering a global network metric less suitable. In addition, regional correlations are sometimes dependent on the number of links included in the networks. For example, we find generally higher (lower) correlations for Europe/North America (Africa) if weaker links are included (excluded), likely because tropical connections have on average stronger dependencies (Supplementary Figs. 9–13).

An interesting question is how to interpret the relationship between precipitation and the causal network skill scores from a physical point of view. Notably, the causal networks are, especially at stringent significance thresholds, dominated by interactions on a timescale of less than 1 week (lag $\tau \le 2$; Fig. 1d). This timescale is broadly equivalent to dynamical interactions related to storm tracks[58]. Simple metrics have been used before to quantify the skill of climate models to capture storm tracks, e.g., pattern correlations in standard deviations of 2–6-days bandpass-filtered daily-mean sea level pressure data[59]. Indeed, Taylor $S$-scores for precipitation are also positively correlated with such simpler metrics (Supplementary Figs. 18–20), which altogether indicates that a large part of the links in the causal networks represent dynamical interactions related to storm tracks. This result is in agreement with earlier work by

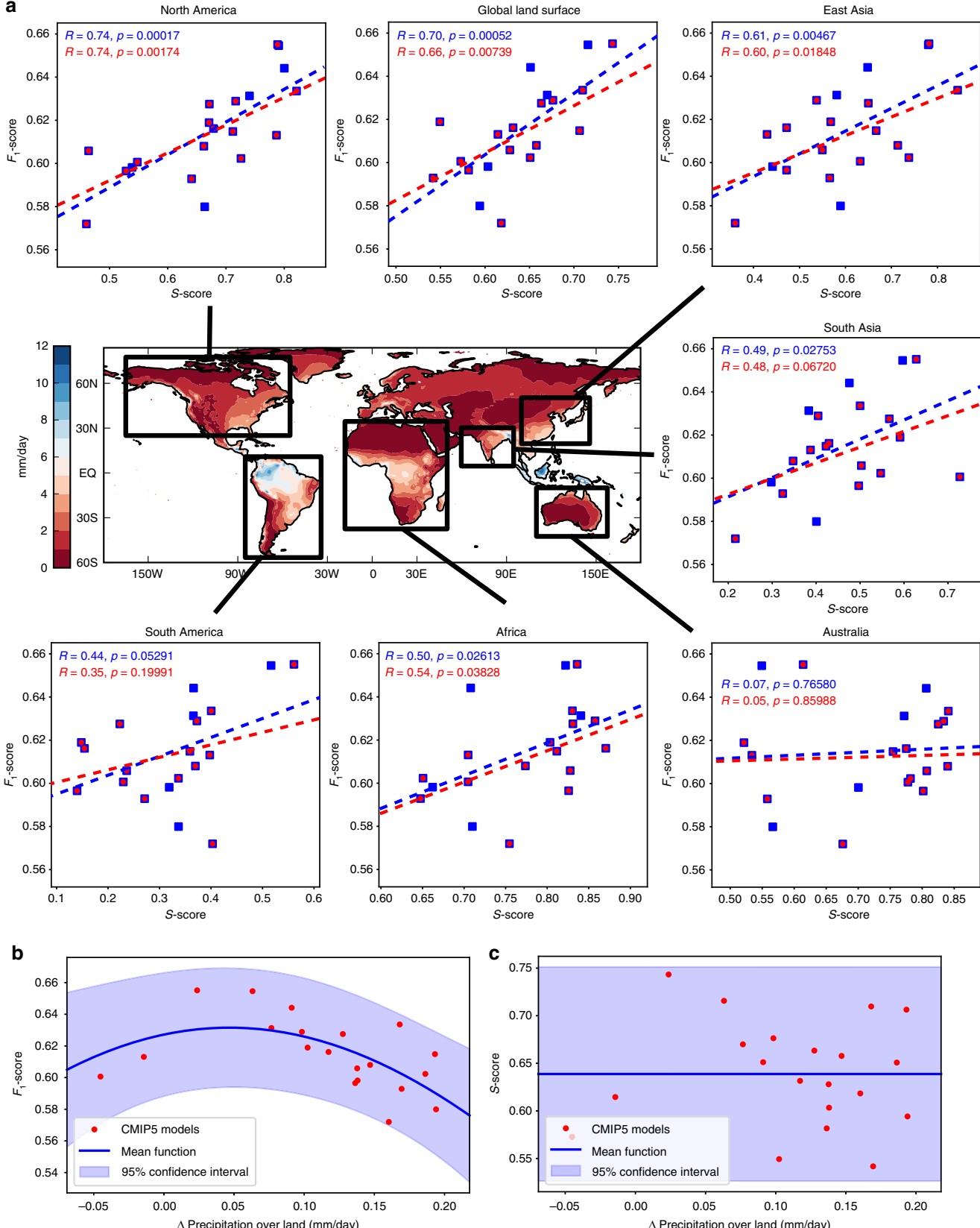

Ebert-Uphoff and Deng[32,33] who constructed networks from DJF and JJA NCEP-NCAR reanalysis geopotential height data, as well as from equivalent data from a single climate model. In their network analyses, they also found storm tracks to be a key driver of network connectivity (see Methods for a comparison of our network methodologies).

Having highlighted the importance of storm tracks, we also point out that the simpler pattern correlation storm track metrics

**Fig. 4 Historical network scores and precipitation. a** Centre map: Climatic Research Unit (CRU) annual mean precipitation rate climatology[57] in mm/day. Surrounding: linear correlations between the $F_1$-scores for the CMIP5 models (with the NCEP-NCAR reanalysis as the reference case) and regional precipitation bias scores ($S$-scores). Higher $S$-scores are equivalent to a better representation of annual mean precipitation in a given model. Correlations are shown for six world regions and for the global land surface (excluding Antarctica), as labelled. Blue denotes data for all models; red the case where five models from causally similar subgroups are excluded (IPSL-CM5A-LR, ACCESS1-3, HadGEM2-CC, NorESM1-M, MPI-ESM-LR). **b** Relationship between $F_1$-scores and land precipitation changes projected by the CMIP5 models. The latter are calculated as the difference between the periods 1860–1910 and 2050–2100 under the RCP8.5 scenario. The relationship exhibits an approximately parabolic structure, as evident from a Gaussian process fit to the data (log-marginal likelihood: 44.15; Methods). Past model precipitation skill as measured through the global $S$-score does not provide a strong relationship (**c;** log-marginal likelihood = 26.35). This result is robust to the use of a different reanalysis, the number of links included in the network, and can also be demonstrated to be statistically significant in a direct parabolic fit (Supplementary Figs. 14 and 15). This implies that precipitation rate changes (Supplementary Fig. 16) can be constrained using the $F_1$-scores (best estimate is around 0.0–0.1 mm/day), whereas past model skill for the same variable does not provide such a constraint; in line with previous demonstrations that past model biases in simple metrics are not necessarily indicative of future model projections[12,76]. Other simple dynamical metrics we tested generally provided lower correlation scores with historical precipitation modelling skill and also did not provide the same emergent relationship for future projections (Supplementary Figs. 17–19). Source data are provided as a Source Data file.

generally show smaller and less significant correlations with the precipitation $S$-scores on a global, as well as on regional scales than our $F_1$-network scores. This underlines that our causal networks identify additional relationships, which further improve the correlations with precipitation. Longer time-scale dynamical interactions, for example triggered by the ENSO and its zonal couplings, as well as its effects on the extratropics are prime candidates for explaining some of the higher skill related to our causal network scores.

Finally, we find strong indications that our causal metrics could aid in constraining uncertainty in precipitation projections under climate change. As mentioned above, past model skill in a quantity does not automatically imply skill for future projections as models can be right for the wrong reasons. The networks here infer rather complex dynamical coupling relationships from sea level pressure data that are effectively impossible to calibrate against current observations, different from, for example, quantities such as global surface temperature[11]. Causal discovery methods could thus provide more robust insights by identifying dynamical coupling mechanisms arising from underlying physical processes, which are more likely to hold also under future climate change scenarios (see Discussion). It is therefore interesting to consider our complex causal information quantity in terms of constraining future precipitation projections. Indeed, we find no relationship between the past global precipitation skill $S$-scores and future precipitation rate changes in the CMIP5 projections, but there appears to be an approximately parabolic relationship between projected CMIP5 global land precipitation rate changes attained by the period 2050–2100 (relative to 1860–1910; Supplementary Fig. 16) and $F_1$-scores from historical runs (Fig. 4b, c). This implies intermediate model range land precipitation changes of around 0.0–0.1 mm/day according to the causal fingerprint scores, as opposed to the most extreme negative and positive changes. We also note that simpler dynamical metrics, e.g., based on sea level pressure Taylor $S$-Scores, or the aforementioned storm track skill scores, and using the same non-parametric Gaussian Process regression (Fig. 4b, c; Methods), do also not yield such emergent relationships (Fig. 4b, c and Supplementary Figs. 17–20).

Any method resting on the assumption that past model skill in a certain metric can be related to projected future changes necessarily suffers from certain restrictions. Firstly, there could be processes that are not at all (or not well) represented in climate models today, which might become important in the future. However, this is true for any emergent relationship based on model evaluation against past observations. Secondly, not all relevant processes might be well-captured through the chosen metric. Our metric here is focused on dynamical processes (although it might, at least indirectly, capture the effects of some

thermodynamical processes[14,60]), whereas, for example, future changes in soil moisture are probably primarily thermodynamically driven. Future changes in soil moisture, in turn, could regionally modulate future changes in land precipitation[61]. Finally, the possibilities for future projections are also constrained by the models participating in CMIP5. Therefore, we can only constrain the relationship within the given data boundaries, and it should be further verified across other scenarios and ensembles (such as CMIP6). Similar model evaluation exercises, also concerning variables other than precipitation and atmospheric dynamical interactions, could test for further emergent relationships in observations and climate modelling projects. Such studies might flexibly combine the blueprint of the method outlined here with other dimension reduction techniques and/or causal discovery algorithms[32,33].

## Discussion

We have highlighted causal model evaluation (CME) as a framework to evaluate state-of-the-art climate models. Based on data-driven causal fingerprints, CME is able to detect models with shared development backgrounds. By considering a large set of climate models simultaneously, we find that climate models with more realistic dynamical causal fingerprints also have smaller precipitation biases globally, and over highly populated areas such as North America, India and China. More realistic fingerprints appear to also have implications for projected future changes in land surface precipitation. Causal network analyses could therefore be a promising tool to constrain climate change projections. The underlying premise is that physical processes (e.g., convection, cloud formation, the large-scale circulation) lead to dynamical coupling mechanisms in Earth's atmosphere. CME aims at statistically representing these couplings in the form causal networks, which in turn are, as we show here, indicative of modelling skill in precipitation. It appears intuitive that modelling skill as captured through our causal fingerprint scores is therefore also relevant for modelling future changes in precipitation, at least so far as the physical processes relevant for present-day precipitation remain important in future climates.

Our work builds on several previous causal network studies in climate science, which were typically focused on network algorithm applications to individual climate modelling or reanalysis data sets, or on the evaluation of dynamical interactions within individual climate models (e.g., refs. [27,32,33,62]). Our results also add to work on global patterns of precipitation co-organisation[63], suggesting atmospheric dynamical interactions as a key driver of important regional climate model errors. We see great scope in using our framework to better understand differences between models and observations, or among climate models, especially

regarding causal interdependencies[26]. Finally, we hope that our work will stimulate the use of novel model evaluation metrics. Causal discovery algorithms have the potential to be at the forefront of this effort as they are able to detect central features of Earth system dynamics such as the direction and time-lag associated with a global teleconnection, opening the door for more in-depth causal interpretation studies[26]. CME could be used to evaluate many other model systems, or could help tracking the impact of model development over time. Ideally, CME will increasingly complement current evaluation approaches[64] and tools[65], and will help constraining uncertainties in climate change projections[66,67], also for climate variables other than global land surface precipitation (Supplementary Fig. 21). The ever-expanding use and development of machine learning techniques in the scientific community[62,68–71], as well as the upcoming CMIP6[3], will greatly accelerate this movement. As such we consider our work as an important stepping-stone for a range of machine learning and other data-driven methods aimed at improving the state-of-the-art of climate modelling and complex system understanding.

## Methods

***F₁-scores for network comparisons***. The network comparisons are purely based on the existence or non-existence of links in a network relative to a given reference network, assuming a certain statistical significance threshold in the PCMCI method ($\alpha$-level). The resulting true links are typically only a small fraction (3–10%; depending on the $\alpha$-level) of all possible lagged connections ($N \cdot (N-1) \cdot \tau_{max} = 24{,}500$) so that the binary (link vs. no link) network comparison becomes an imbalanced classification problem. The $F_1$-score is a widely used, however necessarily imperfect[72], metric for such problems. It balances the statistical precision ($P$) and recall ($R$). It is defined by

$$F_1 = \frac{2 \cdot P \cdot R}{P + R} \qquad (1)$$

With precision and recall defined by

$$P = \frac{TP}{TP + FP} \qquad (2)$$

$$R = \frac{TP}{TP + FN} \qquad (3)$$

where $FP$ ($FN$) is the number of falsely detected links (not detected links) relative to the reference model and $TP$ the number of true-positive detected links. We further modified the definition of the $F_1$-score slightly to account for the sign of dependence (positive or negative) and the networks' discrete time-step nature and the expected natural variance in the precise timing of connections: assuming a link exists in the reference network A, we tested if a matching link with the same sign of dependence exists in network B (with the same causal direction) in a time interval of up to ±2 time lags; equivalent to a time precision of about ± 1 week (6 days). If a link was found at a time-lag not identical with the reference case, the sign of dependence was tested at the original time-step. If also found identical, the link was considered to exist in both networks. Owing to this relaxation of the time-lag constraint, pair-wise network comparison scores do depend on which network is considered as the reference case. As a result, the scores for pair-wise network comparisons shown in Figs. 2a and 3a are not symmetric (cross-diagonal entries are not identical), leading to a larger number of possible comparisons. $F_1$-scores can be calculated for each season, e.g., DJF as shown in Fig. 2. For the historical networks (Fig. 3), an average $F_1$-score was calculated from the individual scores for each of the four seasons as

$$F_1 = \frac{F_{1,\text{DJF}} + F_{1,\text{MAM}} + F_{1,\text{JJA}} + F_{1,\text{SON}}}{4} \qquad (4)$$

***S-scores for measuring precipitation modelling skill***. First suggested by Taylor[56], the $S$-score measures how well a model captures the behaviour of a given climate variable (e.g., temperature, precipitation) over a specific spatial domain relative to an observational data set. It is defined by

$$S = \frac{(1 + R)^4}{4\left(\text{SDR} + \frac{1}{\text{SDR}}\right)^2} \qquad (5)$$

where $R$ is the pattern correlation coefficient between the models and observations and SDR is the ratio of spatial standard deviations between models and observations[55,56]. The calculation of $R$ and SDR incorporate grid-cell area specific

weighting with weights $w$

$$R = \frac{1}{W} \frac{\sum_{i=1}^n w_i \left(x_i - \frac{1}{W}\sum_{j=1}^n w_j x_j\right)\left(y_i - \frac{1}{W}\sum_{j=1}^n w_j y_j\right)}{\sigma_{\text{model}} \sigma_{\text{ref}}} \qquad (6)$$

where $x_i$ and $y_i$ are values for the same quantity (e.g., precipitation rate; mm/day) in a given grid-cell $i$ in the two data sets to be compared, $n$ is the number of grid cells, and $W$ is the sum of area weights

$$W = \sum_{j=1}^n w_j \qquad (7)$$

The spatially weighted standard deviations $\sigma$ (that is $\sigma_{\text{model}}$ and $\sigma_{\text{ref}}$) and the final SDR term are calculated through

$$\sigma^2 = \frac{1}{W}\sum_{i=1}^n w_i \left(x_i - \frac{1}{W}\sum_{j=1}^n w_j x_j\right)^2 \qquad (8)$$

$$\text{SDR} = \frac{\sigma_{\text{model}}}{\sigma_{\text{ref}}} \qquad (9)$$

The $S$-score thus considers both the pattern similarity of a variable over a spatial domain as well as the amplitude ratio, because both spatial coherence and magnitude range are important for measuring model skill[56].

***PCA varimax***. The dimension reduction step (Fig. 1b) serves as a data-driven method to extract large-scale patterns of regional sea level pressure variability that in many cases resemble well-known climatological processes such as the ENSO or the North Atlantic Oscillation (NAO). To extract climatological processes, we here choose truncated principal component analysis, followed by a Varimax rotation (PCA-Varimax)[34,35]. Principal components, often referred to as empirical orthogonal functions (EOFs) in climate science and meteorology, are frequently used to identify orthogonal, uncorrelated global modes of climate variability[25,28,36,37]. To remove noisy components, we then truncate and keep only the first 100 leading components in terms of their explained variance. The additional Varimax rotation on these leading components then maximises the sum of the variances of the squared weights so that the loading of weights at different grid locations will be either large or very small. It has been shown that this leads to more physically consistent representations of actual climate modes, mainly because the Varimax rotation allows spatial patterns associated with the components to become more localised and their time series of weights to be correlated, as is the case for actual physical modes[25,36,37]. Principal components without rotation consecutively maximise variance and therefore often mix contributions of physically defined modes such ENSO, Pacific Decadal Oscillation (PDO), or the NAO, whose time-behaviour is not orthogonal, making patterns more difficult to interpret. We here estimated the spatial pattern (loading) of the Varimax components from 70-year (1948–2017) daily sea level pressure anomalies of the NCEP-NCAR reanalysis data set[52], and then used these weights to also consistently extract the Varimax component time series from the CMIP5 sea level pressure simulations. The motivation behind using sea level pressure as the variable underlying the networks is that it is a standard variable to characterise large-scale atmospheric dynamics and corresponding variability, e.g., in climate modes or weather patterns. Therefore, it is also available in virtually any reanalysis data set or model data archive, which allowed us to work with the largest possible number of ensemble members for the CMIP5 analysis. The components obtained for the four meteorological seasons for the NCEP-NCAR data can be found in Supplementary Figs. 22–42[1]. For the subsequent causal discovery method, we further filtered weights in terms of their spatial separability and their frequency spectra, leading to a total of 50 components for each season. For example, we typically excluded components that exhibited a sudden change in behaviour when entering the satellite era (1979-), which resulted in unresolved frequency spectra (e.g., DJF components 18, 36, 38, 41 provided as Supplementary Figs. 40, 58, 60 and 63). Such apparently unphysical component time series changes were in particular found in Asia, Africa and the Middle East and could therefore be related to a lack of historical data coverage feeding into the reanalysis in those regions. To further control for the importance of choosing a certain set of components for the overall results and conclusions, we sometimes included some of these components for certain seasons (e.g., component 7 for DJF), but we did not find any noticeable sensitivity of the relative $F_1$-scores to this selection process. A side effect of this selection process, however, remains a reduced network coverage in those areas. Overall, we found that the global network metrics were effectively insensitive to the choice of nodes and their geographical distribution. This is also evident from the relative insensitivity of the model rankings to the specific season (Supplementary Figs. 1, 3–6 and Supplementary Table 1). The indices of the 50 components chosen for each season are provided in Supplementary Table 3. The component time series were averaged to 3-day-means before the application of PCMCI. This time-aggregation presents a compromise to resolve short-term interactions in our intercomparison (a few days), while limiting the increase in dimensionality due to additional time lags (here, 10 time lags for $\tau_{max}$, i.e. 30 days).

**PCMCI causal discovery method**. PCMCI is a time series causal discovery method further described in reference[23]. Commonly, causal discovery for time series is conducted with Granger causality, which is based on fitting a multivariate autoregressive time series model of a variable $Y$ on its own past, the past of a potential driver $X$, and all the remaining variables' past (up to some maximum time delay $\tau_{max}$). Then $X$ Granger-causes $Y$ if any of the coefficients corresponding to different time lags of $X$ is non-zero (typically tested by an $F$-test). As analysed in reference[23], Granger causality, due to a too high-model complexity given finite sample size, has low detection power for causal links (true-positive rate) if too many variables are used and for strong autocorrelation, both of which are relevant in our analysis. PCMCI avoids conditioning on all variables by an efficient condition-selection step (PC) that iteratively performs conditional independence tests to identify the typically few relevant necessary conditions. In a second step, this much smaller set of conditions is used in the momentary conditional independence (MCI) test that alleviates the problem of strong autocorrelation. In general, both the PC and MCI step can be implemented with linear or nonlinear conditional independence tests. Here, we focus on the linear case and utilise partial correlation (ParCorr). A causal interpretation rests on a number of standard assumptions of causal discovery as discussed in reference[22], such as the Causal Markov assumption, Faithfulness, and stationarity of the causal network over the time sample considered. The free parameter of PCMCI is the maximum time delay $\tau_{max}$ here chosen to include atmospheric timescales over which we expect dependencies to be stationary. The pruning hyper-parameter pc-$\alpha$ in the PC condition-selection step is optimised using the Akaike information criterion (among pc-$\alpha$ = 0.05, 0.1, 0.2, 0.3, 0.4, 0.5). PCMCI yields a $p$-value (based on a two-sided $t$-test) for every pair of components at different lags. We defined links in the networks using a strict significance level of $10^{-4}$ in the main paper. However, very similar results are found for other more relaxed or even stricter significance levels; as demonstrated extensively in the Supplementary Material.

**Other network construction methods**. As discussed in the main text, causal networks have been used several times before in the climate context. Two of the most prominent cases of such studies are those described in references[32,33], where Ebert-Uphoff and Deng also discuss remote impacts and information pathways, as well as the role of storm tracks as important drivers of network connectivity. Their work is further a good demonstration of other possible ways to construct causal networks, the effect of which might be an interesting topic for future studies. For example, their network approach was carried out on a grid-cell-wise level rather than using PCA Varimax components. The latter are designed to capture distinct regional climatological processes while an analysis at the grid-cell level is more granular which, however, carries the challenges of higher dimensionality, will have a strong redundancy among neighbouring grid cells, and grid-level metrics will require handling varying spatial resolution among data sets. Furthermore, the original PC causal discovery algorithm used in their work is less suited for the time series case than PCMCI[23]. They also used another meteorological variable (500 hPa geopotential height) to construct their networks and compared aggregate network metrics rather than comparing networks on a link-by-link basis.

**CMIP5 data**. For the network constructions, we used daily-mean sea level pressure data from the CMIP5 data archive, as stored by the British Atmospheric Data Centre (BADC). An overview of all models and simulations used is given in Supplementary Table 2. The 20 models used for the pre-industrial networks are as labelled in Fig. 2a. The 20 models used for the historical and RCP8.5 reference case are as labelled in Fig. 3a. Typically, we used the final 210 years of each pre-industrial simulation, assuming that these years represent the most equilibrated state of each model. For historical and RCP8.5 simulations, we used at least three ensemble members, which typically covered 70 years between 1st January 1936 and 31st December 2017. Relaxing the left time boundary by up to 12 years relative to the reanalysis data time period allowed us to include more models, as some modelling centres ran more historical than RCP8.5 simulations. If sufficient data was available for both the historical and RCP8.5 simulation, the two simulations were merged on 1st January 2006; the day after historical simulations ended in most cases. All data (including the reanalysis data sets) was linearly de-trended on a grid-cell basis and seasonally anomalized by removing the long-term daily-mean. Note that sea level pressure data is effectively stationary even under historically forced climatic conditions so that the de-trending is a prudent step to remove any potentially occurring small trends to a good approximate degree. Of course, we cannot fully account for the very long time-scales that may be associated with some climate processes[73] beyond the time-scale covered by each individual data set. Each model data set was bi-linearly interpolated to a 2.5° latitude x 2.5° longitude grid in order to extract the component time series based on the Varimax loading weights computed from the NCEP-NCAR[52] reanalysis data.

**Precipitation data**. As observational reference, we used the land surface CRU TS v4.02 data set from the University of East Anglia[57], which does not cover Antarctica. CMIP5 precipitation data was taken from single ensemble members (Supplementary Table 2) of the historical and RCP8.5 simulations, as described above. As for the sea level pressure data, all precipitation data was bi-linearly interpolated to the NCEP-NCAR spatial grid prior to the intercomparison. Climate change-induced differences shown in Fig. 4b, c were calculated by subtracting the model-specific land surface (using an ocean and Antarctica mask equivalent to the one of the CRU data set) average precipitation rate for the period 1860–1910 (covered by all models) from the same measure for the years 2050–2100.

**Random and Independent data**. The data sets for the Random and Independent case in Fig. 2a were created with Gaussian noise driven multivariate autoregressive models of the same number of variables as in the original data. For the Independent case only the lag-1 autocorrelation coefficients are non-zero and set to a value of 0.7. Hence, all variables are independent, but due to finite sample effects, the estimated networks with PCMCI will still contain some cross-links. For the Random case, we created a random network with a link density of 5%, randomly connecting two components at lag-1 with a coefficient of 0.1, in addition to autocorrelation coefficients with a value of 0.7 for each component. Like for the original data, we simulated three data sets (covering 70-year periods of the 210 years) with the same sample size as the original data.

**Gaussian process regression**. To estimate the nonlinear dependency between $F_1$/$S$-scores and land precipitation changes (Fig. 4b, c and Supplementary Fig. 14), we used Gaussian Processes (GP) as a widely used Bayesian non-parametric regression approach[74]. We implemented the GP with a standard radial basis function kernel with an added white noise kernel and optimised the hyperparameters using the log-marginal likelihood. The resulting fit line is approximately parabolic when using the $F_1$-score. In Supplementary Fig. 15 we also directly fit a parabolic function $y = a + bx + cx^2$.

## Data availability

All raw sea level pressure, surface temperature and precipitation rate data is publicly available. CMIP5 data is available through the Lawrence Livermore Laboratory (https://pcmdi.llnl.gov/mips/cmip5/availability.html) and many other sources such as the British Atmospheric Data Centre (BADC, http://www.badc.rl.ac.uk/) as variables 'psl', 'tas' and 'pr', see Supplementary Table 2 for an overview of all selected simulations. CRU precipitation rate data is publicly available through, e.g., https://crudata.uea.ac.uk/cru/data/hrg/; as is the NCEP-NCAR reanalysis through https://www.esrl.noaa.gov/psd/data/gridded/data.ncep.reanalysis.html. ERA-Interim data is accessible via https://www.ecmwf.int/en/forecasts/datasets/reanalysis-datasets/era-interim. The source data underlying Figs. 2a–f, 3a, b and 4a–c are provided as a Source Data file.

## Code availability

Tigramite source code is available through https://github.com/jakobrunge/tigramite. Example Jupyter-notebooks and Python code used to carry out the Varimax and PCMCI analysis here will be made available through https://github.com/peernow/CME_NCOMMS_2020.

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

## Acknowledgements

P.N. is supported through an Imperial College Research Fellowship. J.R. was supported by a Fellowship from the James S. McDonnell Foundation. We acknowledge the World Climate Research Programme's Working Group on Coupled Modelling, which is responsible for CMIP, and we thank the climate modelling groups (listed in Supplementary Table 2 of this paper) for producing and making available their model output. For CMIP5, the U.S. Department of Energy's Programme for Climate Model Diagnosis and Intercomparison provides coordinating support and led development of software infrastructure in partnership with the Global Organisation for Earth System Science Portals. For plotting, we used Matplotlib, a 2D graphics environment for the Python programming language developed by J.D. Hunter. For causal discovery we used the Tigramite package (version 4.1) available from https://github.com/jakobrunge/tigramite. We used the JASMIN postprocessing system provided through the Centre for Environmental Data Analysis (CEDA)[75]. We thank James King (University of Oxford) for helpful discussions.

## Author contributions

P.N. and J.R. together suggested and designed the study. P.N. led the scientific analysis and paper writing in collaboration with J.R. All authors (i.e., P.N., J.R., V.E. and J.D.H) contributed to the scientific interpretation of the results and to the paper writing.

## Competing interests

The authors declare no competing interests.
