## [Peer Review File · Nature Communications]

Reviewers' comments:

Reviewer #1 (Remarks to the Author):

A review of "Causal teleconnection fingerprints in observations and climate models"

Tamas Bodai

The authors apply a set of algorithms to establish a teleconnection characteristic based on climate data. It starts with a dimension reduction technique, an advanced version of EOF analysis, which finds spatially distinct localised modes of variability. These locales will represent nodes of a teleconnection network. (I suppose different choices of an observable climatic quantity will establish different network nodes. The authors choose to consider -- for an unspecified reason -- the sea level pressure field.) Then the edges of the network are established such that they represent Granger causality. (The authors make a point to represent properly what is an "indirect effect" or a "common cause effect".) This network the authors call a "causal teleconnection fingerprint". (Although I don't like this expression because the word "fingerprint" suggests that there is something real behind it, the "finger" that "leaves the fingerprint", whereas in fact it's an emergent characteristic of a model, which is different in every model (world).) The authors run their algorithms to establish the teleconnection characteristics of 20 something CMIP5 models, on the one hand, and some reanalysis data product as the "truth" or the "observation", on the other hand. They then can measure the difference/"distance" F1 between pairs of models or between a model and the "observation". There are several interesting findings regarding the variation/diversity of the model teleconnection characteristics. They are able to verify if some models share a developmental history and so "structural commonalities" as they result in similarities of emergent characteristics. (Of course, the latter is not a necessity, and so the "structural commonality" won't be detected if it does not result in similar emergent characteristics.) However, the usefulness of this is not demonstrated. Furthermore, by parting the data for each model, they show that data from the same model establish the teleconnection characteristic closely enough that they can be told apart from other models. On the one hand, this is a trivial finding, to do with data availability rather than the models or the algorithm; therefore it should be viewed just as a necessary checking of the results. On the other hand, it's a disappointing finding, or perhaps rather echoing of what we already know, namely, that the models are quite different. Secondly, what seems to be the main application of the knowledge of the "teleconnection characteristic", the authors establish that models that are more similar in this regard to the truth (smaller F1), make smaller errors in global and some regional precipitation climatology. (This is established as a correlation.) They claim that this finding can help constrain the uncertainty in precipitation projections. With some caveats, this seems an important and maybe useful result.

I don't have major concerns with the paper. It was a worthwhile read, and clearly a great effort. It's a paper that reports on findings that are not interpreted or explained, but I don't mind that. Some part

of the methodology at least should be either explained or justified. I attach an annotated version of the manuscript with comments on such and other issues.

Note: My policy is to not make a recommendation to editors on the publication of manuscripts. It is the editor's duty to make up their mind based on (ideally factual) referee reports, or one that reflects the referee's (ideally unbiased) opinion.

Reviewer #2 (Remarks to the Author):

This paper reports findings of an analysis where causal discovery algorithms were applied to identify causal fingerprints in both observations (reanalysis products) and CMIP5 model simulations. The authors conclude that a model's capability of simulating precipitation patterns over selected regions highly correlates with the model's skill of capturing these causal fingerprints. I agree fully with the authors that machine learning has huge potential for contributing to the study of atmospheric processes given the enormous amounts of data available from routine meteorological observations and weather and climate model simulations.

However, I have to recommend a rejection of the present manuscript due to the following: "causal teleconnection fingerprints" are stated as the subject of the study yet the authors clearly misunderstood the concept of teleconnection in atmospheric sciences. Teleconnection refers to synchronized changes in monthly (or longer time) mean atmospheric fields (e.g., temperature, pressure or geopotential height) over different geographical locations. Classic examples include the North Atlantic Oscillation (NAO) and Pacific North American pattern (PNA). The 3-day averaged SLP data, after the further dimension reduction via PCA, contains mainly synoptic and sub-monthly-scale variability that are effectively "weather", not classic modes of teleconnection. This explains why the PCA patterns shown in the supplementary plots are all spatially isolated pressure anomalies. Due to the weather, thus short-lived nature of these anomalies, the causal connections among the nodes based upon PCA components are very sparse except for regions where background steering flow is prominent enough to allow upstream weather to cause changes in the weather downstream. This also explains why the time lags identified for these causal connections are mostly 1s (i.e., 3 days) with a few 2s. Since the causal fingerprints detected here are physically weather anomalies that are occasionally connected over a short timescale, there exist much simpler dynamics-based metrics for evaluating model's simulations of these features. For example, one can classify the CMIP5 models simply by calculating the spatial correlations between observed and modeled seasonal mean winds and storm tracks (e.g., in terms of variance of band-passed pressure or height fields). Model's skills of simulating the seasonal mean winds plus storm tracks are highly correlated with model's simulation of precipitation. These metrics are much easier to calculate compared to the causal fingerprints discussed in this paper.

In summary, the authors misinterpreted the identified causal fingerprints as those associated with teleconnections while in fact those are sparse connections between local weather anomalies. Much

simpler metrics can be constructed based on the physical nature of these anomalies to classify and evaluate model performances in simulating critical fields such as precipitation. The present manuscript fails to convince this reviewer that we gain anything new by going through this practice. There have also been studies in the past discussing causal connections among atmospheric weather disturbances across daily timescales thus the application itself is not innovative either.

26th November 2019

Nature Communications Manuscript NCOMMS-19-25061 - Reply to the Referees

Below we reply to each of the two referee's comments (*italic*) point-by-point in **bold** font.

Reply to Referee 1:

We thank Dr. Bodai for his overall positive and constructive review, which has helped us to improve our manuscript significantly.

The authors apply a set of algorithms to establish a teleconnection characteristic based on climate data. It starts with a dimension reduction technique, an advanced version of EOF analysis, which finds spatially distinct localised modes of variability. These locales will represent nodes of a teleconnection network. (I suppose different choices of an observable climatic quantity will establish different network nodes. The authors choose to consider -- for an unspecified reason -- the sea level pressure field.) Then the edges of the network are established such that they represent Granger causality. (The authors make a point to represent properly what is an "indirect effect" or a "common cause effect".) This network the authors call a "causal teleconnection fingerprint". (Although i don't like this expression because the word "fingerprint" suggests that there is something real behind it, the "finger" that "leaves the fingerprint", whereas in fact it's an emergent characteristic of a model, which is different in every model (world).)

We have now included a motivation for why we use sea level pressure in the Methods section, see our reply to another comment below.

Concerning the use of the term 'fingerprint', we also reply in detail to a similar comment below.

The authors run their algorithms to establish the teleconnection characteristics of 20 something CMIP5 models, on the one hand, and some reanalysis data product as the "truth" or the "observation", on the other hand. They then can measure the difference/"distance" F1 between pairs of models or between a model and the "observation". There are several interesting findings regarding the variation/diversity of the model teleconnection characteristics. They are able to verify if some models share a developmental history and so "structural commonalities" as they result in similarities of emergent characteristics. (Of course, the latter is not a necessity, and so the "structural commonality" won't be detected if it does not result in similar emergent characteristics.) However, the usefulness of this is not demonstrated. Furthermore, by parting the data for each model, they show that data from the same model establish the teleconnection characteristic closely enough that they can be told apart from other models. On the one hand, this is a trivial finding, to do with data availability rather than the models or the algorithm; therefore it should be viewed just as a necessary checking of the results. On the other hand, it's a disappointing finding, or perhaps rather echoing of what we already know, namely, that the models are quite different. Secondly, what seems to be the main application of the knowledge of the "teleconnection characteristic", the authors establish that models that are more similar in this regard to the truth (smaller F1), make smaller errors in global and some regional precipitation climatology. (This is established as a correlation.) They claim that this finding can help constrain the uncertainty in precipitation projections. With some caveats, this seems an important and maybe useful result.

Dr. Bodai acknowledges that our manuscript presents several interesting results, but also re-considers if they are trivial. We further clarify below why these findings are important.

First of all, he acknowledges that our novel causal model evaluation method works; it can detect similar datasets as similar, has implications for other key climate variables such as precipitation and

also provides emergent relationships for future precipitation projections. To provide further evidence for this last point, in our revised manuscript we show that other simple metrics such as past precipitation modelling skill, or storm track skill metrics are not able to provide such constraints.

What might appear to be a trivial finding as a result, i.e. we can use the method to show that models are different, is indeed not trivial. The algorithm clearly filters out causal structures that can be shared or different among models/observations. As the reviewer acknowledges, this finding was a crucial requirement to ensure that the algorithm has validity and works on large-scale dynamical interactions in terms of model evaluation and intercomparison. Indeed, what we describe in our paper is the first demonstration that such algorithms could be useful for this purpose. Identifying interdependencies among the CMIP models has been identified as one of the main research needs. From Eyring et al., Nature CC (2019): "In addition, it has been demonstrated that CMIP models are not independent. Most inferences in the literature about model interdependence are derived from error correlation. This cannot identify the specific model components that are interdependent. Identification of these common components is a difficult task due to the large number of models involved in CMIP and lack of detailed information regarding individual model versions. Further work is required to understand how interdependence can be best assessed."

We therefore cannot echo the concern by Dr. Bodai that these findings might be disappointing in the sense that models are different to each other and to observations (i.e. imperfect), and following this discussion we would hope that he agrees with us. Quite to the contrary, one could actually argue that different models are required in order to span the uncertainty that is introduced for example because of different parametrizations in the models.

I don't have major concerns with the paper. It was a worthwhile read, and clearly a great effort. It's a paper that reports on findings that are not interpreted or explained, but i don't mind that. Some part of the methodology at least should be either explained or justified. I attach an annotated version of the manuscript with comments on such and other issues.

Thank you for this positive conclusion. We have taken up all your suggestions and reply to them point-by-point below. In particular, this has helped us to expand on the interpretation and explanation of the results. In addition, we have added additional explanations concerning the physical interpretation of our results, see our response to another comment below.

page1:

'reliable'. I think we should know already in the abstract what makes the method reliable. This word suggests that the method is somehow encompasses all possible viewpoints, and so no complementary analysis is possible or needed. I wouldn't think that you want to suggest that.

Fair point; the aspect of reliability obviously arises from our analysis itself, for example that we can reliably detect similar datasets as similar. Since we explain those points anyway further below, we just removed the word 'reliable' here to avoid confusion.

How is objective different from reliable?

This point has, as part of our previous response, already been addressed.

I don't think that comparing a model to other wrong models constitutes "benchmarking". The word benchmarking means I think that your reference for the comparison is undisputed, as only then you would know what error your benchmarked model makes. Other models are not something like this. Are observations good references for benchmarking? Let's not discuss now the "evolution" of "observational data products" through different versions. Observations represent only a single realisation of a chaotic system. A historical uninterrupted simulation of a climate models is also such a single realisation. However, other realisations might behave very differently, i.e. the system displays internal variability. Only if this internal variability is very small regarding the observable and associated effect of interest can observations possibly serve as a reference for benchmarking.

We changed the wording to 'evaluate'. We also note that our CME framework aims to achieve exactly this, i.e. to extract a significant pattern of interactions in the presence of interfering noise.

'using machine learning': I came back here after learning the methods section, and i didn't read about machine learning there. At least not explicitly. Apologies if i overlooked some relevant reference in this regard.

In this last part of the abstract, we just meant to mention the broadest possible implications: we start with a rather large dataset of spatio-temporal sea level pressure variability and then apply dimension-reduction methods (unsupervised learning) and afterwards high-dimensional (linear) regression (supervised learning) to this data. In the broadest sense this would certainly qualify as machine learning as everything is entirely data-driven and relationships/patterns are learned purely by the algorithm. We use two quite established algorithms to carry out these two steps (PCA Varimax and partial correlation) but these could well be replaced by more complex algorithms, e.g. Gaussian Process regression for the later step (cf. Runge et al. 2019), in future applications of this novel type of model evaluation method. Without a doubt it is a data-driven method. Therefore, our generic statement on 'using machine learning and other data-driven methods'. However, in response to your comment we have removed this sentence from the abstract.

page 2:

'development-specific choices'. I am not sure what this means

This refers to the fact that climate modellers always face choices when developing new climate models, e.g. which processes to represent in a given model set-up and how (e.g. different ways to parameterize convection, clouds etc). We have thus revised this sentence to:

"However, while all climate models are based on the same physical principles, there are development-specific choices that lead to significant model differences, in particular related to subgrid-scale parameterizations of clouds, convection and aerosols¹⁻³."

'persistent uncertainties'. One might think this is some statistical term.

In a way, we indeed refer to statistical quantities here; just that there are different uncertainties related to different quantities, research questions etc. As Dr. Bodai points out himself above, there is no perfect ground truth/benchmark so that even model disagreements with observations are subject to uncertainties. To avoid confusion, we have rephrased the sentence to:

"These contribute to persistent discrepancies between models and observations as well as among model projections, for example regarding precipitation changes⁴⁻⁶."

'novel approach': What is really the new contribution in this paper? In the methods section there seems to be citation for all parts of methodology.

The algorithms have been developed elsewhere. The new contributions are:

- 1) We apply this recently developed causal discovery algorithm for the first time to a large ensemble of climate models, introducing an entirely new approach for climate model evaluation.**
- 2) We show for the first time that this algorithm provides reasonable results when used in this context, i.e. for causal model evaluation (CME). We hope, of course, that our paper is just a starting point for a number of studies looking into how causal information measures can help us evaluate models and find emergent constraints.**
- 3) We show for the first time that the algorithm works for these large sets of global climate data, i.e. that it can detect similar causal dependency structures between sets of models and identify similar models as such.**
- 4) We show for the first time that the resulting causal skill metric has meaningful implications for how well a model describes other key climate variables such as precipitation.**
- 5) We show for the first time that these causal fingerprints provide emergent relationships that can be used to put constraints on future climate change projections, and furthermore that simpler skill metrics do not achieve this goal.**

We have revised the wording of our paper title, abstract, and several other parts of our manuscript, to better highlight these various novelty aspects.

'Below, we show that, first, characteristic causal teleconnection fingerprints can be learned from climate datasets.': I don't understand what you mean. You find causal teleconnections in the data because they are there and you have enough data for detection. But is it really so unexpected? Has no-one ever found causal teleconnections in climate data yet? You should just say that you mapped these connections out. This shouldn't be a numbered item of conclusions or take-home messages.

'Second, these fingerprints are robust among ensemble members of the same model and, for example, can identify shared development backgrounds.': This is what you find. The conclusion is that you had long enough time series for each ensemble member (and also that non-stationarity/climate change didn't have much effect); no? Of course, in one model, under stationary climate, with infinite data, there would be zero variability among ensemble members. Again should this really be a numbered outcome/conclusion?

We think this is a misunderstanding. It is not meant to be a summary of conclusions, but a quick run through the steps taken in the paper as is commonly done in paper introductions. Of course, we expect to learn dynamical interactions if the dataset is long enough. However, as we mentioned above this has never been applied to compare and evaluate climate models on the scale that we have achieved here.

'fingerprints': Overused expression. Also I wouldn't even think that it's the right word. The idea behind the expression fingerprint is that there is something real that left a mark, which is not a full undistorted image. A bit like a finite sample drawn from a distribution. In contrast here you have different models that are genuinely different worlds. There is not an identical teleconnection characteristics/signature that shines through the finite simulation data. The teleconnection signature is an emergent property of the models, it is not put in there.

We understand Dr. Bodai's concern that models are, of course, never real, and even reanalysis datasets used for the evaluation of models are of course an imperfect reflection of reality. Having said that, climate models and reanalysis datasets are indeed trying to model a common physical, chemical and biological climate system, which should have emergent properties for itself. How well, or if, these emergent properties are represented in a given climate dataset from reanalyses or models is the highly interesting question, which we are (to a degree) trying to answer in our manuscript. We are partly using the term fingerprint here as a causal extension of the term as it has been used in climate change attribution science (e.g. Hegerl et al. 1997, 2011). We therefore find the term 'fingerprint' actually quite apt in this context in the sense that a fingerprint is something unique for a given dataset, which will be more similar for datasets from a 'dataset family' such as different types of reanalysis datasets, or datasets produced by models with a similar model development background.

Hegerl et al. Patterns of change: whose fingerprint is seen in global warming, *Environmental Research Letters* 6, 044025 (2011).

Hegerl et al. Multi-fingerprint detection and attribution analysis of greenhouse gas, greenhouse gas-plus-aerosol and solar forced climate change, *Climate Dynamics* 13, 613-634 (1997).

However, in order to consider this comment, we have removed the word "fingerprint" from the title of the manuscript, and use it more sparsely throughout the manuscript.

'understand differences between models and observations': another big promise. In the end you didn't deliver on this.

Causal discovery algorithms were designed for inference, i.e. to better understand interactions in systems. We interpret some of the interactions here in the Supplementary, e.g. in Figure S2, but a detailed comparison of the networks and their differences would clearly be beyond the scope of this paper. We still found it quite important to highlight that our paper opens the door for such studies in the future. We have added a half-sentence on this in the Conclusions of the paper:

"Causal discovery algorithms have the potential to be at the forefront of this effort as they are able to detect central features of Earth system dynamics such as the direction and time-lag associated with a global teleconnection, opening the door for more in-depth causal interpretation studies²⁴."

p. 3:

'by expert knowledge': It appears that it is rather the selection of the physical quantity, the pressure, that is guided by expert knowledge (or perhaps rather guided by the interest of the expert), not the "spatial structures". The latter is selected by the "VARIMAX rotated PCA" algorithm rather "mechanically".

This is true. However, one might also be interested to consider networks between a certain set of climate modes and thus select specific PCA Varimax components (see also our Methods section). It really depends on the science application. We consider our paper as a blueprint for a methodology applying causal discovery algorithms to model intercomparison. Another paper might apply the same workflow to other datasets, e.g. stratospheric circulation anomalies, or many other Earth science features as yet unidentified.

'Supplementary files 2 and 6': Not clear what file is referenced. I've got files named like 218050_0_supp_3966261_pv7c4r.pdf

Thanks for pointing this out. This must be a renaming convention introduced by the Nature Communications submission system. We have added a label to the top lines of each Supplementary file so that they should be easier to identify even if the same issue occurs again.

p. 4:

i guess it's not an easy thing to convey information exactly by a picture, but a "common driver" is a node, while here edges are pointed to.

We agree and therefore have changed the labels to 'spurious links'.

'3-day-averaged': Why?

We took three day-average data as this presents a good compromise between the time-scales we want to resolve (up to 30 days) and the short-term interactions we still want to include (a few days) in our intercomparison. One could also choose daily time resolution for example, but this would lead to a threefold increase in dimensionality and would thus increase the risk of fitting noise due to autocorrelation and the curse of dimensionality (i.e. reduced detection power of the algorithm). We have added the following explanation to the Methods section:

"This time-aggregation presents a compromise to resolve short-term interactions in our intercomparison (a few days), while limiting the increase in dimensionality due to additional time lags (here 10 time lags for $\tau_{\max} = 30$)."

'autocorrelation': Why is it relevant?

We have added a half sentence to the figure caption, highlighting that autocorrelation could be considered as self-directed links to the node itself.

p. 5:

cancel 'that is benchmarked against'

Done.

'400-500 most significant links': among all models?

Given the chosen significance level, every network for every individual climate model or reanalysis dataset has around 400-500 links. To clarify this point we revised the sentence to:

"The network comparison results depend on the number of links considered to be statistically significant (Methods). However, we tested that all conclusions based on the 400-500 most significant links per network included here are robust to a large range of possible network link densities from a hundred to more than a thousand links (Supplementary Figures S3-S6; Table S1)."

'Each model can be detected individually': I understand that "detect" here has a specific disciplinary meaning, but still i would try to word this sentence differently, because it sounds weird -- as if we wanted to play a game of "which model am i". Also, avoid making it sounds that this is about either the model or the methodology. It is about data set size. And so, as such, it's not a so important point. This is important only to show confidence (or statistical figures) about your results.

Thank you, we changed the wording to 'recognize'.

p. 6:

'Climate models are recognized to share a physical ground truth': I wouldn't think that we need to "go back to square one". I'm pretty sure there is plenty of good arguments and evidence for that climate models have something to do with the climate. Imagine what this would tell a "(man-made) climate change denier".

Point taken, but we really just re-emphasize that the models share physics and are not behaving randomly, so this should rather contradict any climate change denier, we would hope! The result also underlines that our algorithm is not just picking up on random relationships in the data.

p. 7:

'each individual network comparison score': Individual with respect to what? I just note that i came to this figure from the Methods section (paragraph for F1)!! This is how i ended up here the first time the most naturally. How many coloured marker are there and why?

Individual means for each network. In (b) to (f) the comparison scores are those relative to the reference networks (stated in the title of each subfigure). The idea behind the paper's structure is to provide first an overall view of the topic at hand and subsequently the Methods section on details of how the calculations were done for the interested reader. Data points for individual models are marked by different colours, where we used the same colour, however, for models for which the data was contributed to CMIP5 by exactly the same modelling centre (using different model versions).

I don't really understand why a model doesn't agree with itself. Is there not an absolute top F1 score for complete agreement, even if that's based on finite data? (The red crosses are at different levels; and in panel (a) the diagonal is nonuniform.)

Teleconnections/atmospheric dynamical interactions can have long time-scales of variation, which is why it was so important for us to show for the first time that we have a sufficient data record here to detect their signal. If each ensemble member of a given model would have infinite sample size, the F_1 -score between them would converge towards 1. Finite data leads to differences due to (1) long time-scales of variation in interactions and (2) variance of the estimator of the networks, i.e. PCMCI. The difference from the perfect score on the diagonal of the comparison matrices is due to these two and will also be model-dependent, e.g. due to the signal-to-noise ratio for a given model system.

p. 8:

Figure 3a: Why do you use a different range for the colours wrt. Fig. 2 a? It would be good to see if the teleconnection characteristics change under external forcing. Can we say if any change is significant? You only tested for a difference between models.

In Figure 2a, we included random and independent networks so that we had to use a different colour-scale due to the very low associated F_1 -scores. In Figure 3a we want to give a better impression of the climate model differences/similarities, which is only possible with an adapted colour scale. It would be interesting to see how much networks for the same models have changed from pre-industrial to the historical simulations and we intend this as the subject of a future study, where sufficient

emphasis could be put on this aspect. However, it is beyond the scope of this study and might distract attention from our main messages.

I don't understand what differences you are referring to. What is the input for the Kruskal-Wallis test?

As inputs we provide here each model group of points in Figure 3b, with each group representing a separate distribution. We test for differences in the medians of these distributions. See also this Python documentation for the function used for the calculation:

<https://docs.scipy.org/doc/scipy/reference/generated/scipy.stats.kruskal.html>

'biases': I'm not sure you use this term correctly here. I think you simply mean errors -- in the sense of the word in "root mean square errors". Bias is an expected/systematic error.

Thanks for pointing out. We have adapted the wording accordingly.

p. 9:

It's interesting that you consider teleconnection between the pressure in different regions, yet it has something to do with precipitation. Do you have an explanation for this?

Yes, there is indeed a plausible physical connection. Teleconnections are large-scale control mechanisms on weather, including precipitation. Therefore, it is indeed reassuring that the networks we learn can give us an impression on how well models also capture global precipitation patterns. In addition, we have added the following paragraph to the manuscript, partly in response to the second referee:

"An interesting question is how to interpret the relationship between precipitation and the causal network skill scores from a physical point of view. Notably, the causal networks are, especially at stringent significance thresholds, dominated by interactions on a timescale of less than one week ($\text{lag } \tau \leq 2$; Figure 1d). This timescale is broadly equivalent to dynamical interactions related to storm tracks⁵⁶. Simple metrics have been used before to quantify the skill of climate models to capture storm tracks, e.g. pattern correlations in standard deviations of 2-6-days bandpass-filtered daily mean SLP data⁵⁷. Indeed, Taylor S-scores for precipitation are also positively correlated with such simpler metrics (Supplementary Figure S18-S20), which altogether indicates that a large part of the links in the causal networks represent dynamical interactions related to storm tracks. At the same time, these simpler metrics generally show smaller and less significant correlations with the precipitation scores on a global as well as on regional scales, highlighting that our causal networks identify additional relationships which further improve the correlations with precipitation. Longer time-scale dynamical interactions, for example triggered by the ENSO and its zonal couplings as well as effects on the extratropics are prime candidates for explaining some of the higher skill related to our causal network scores."

I think the existence of a correlation would also (or rather?) require an explanation. This goes back to my point that your choice of the pressure is rather an expert choice (and of course it's science, not black magic, so it has to be explicable, verifiable). Perhaps it's already done in the literature; but then cite it.

Just in the paragraph above we motivate this aspect by saying and citing:

"Teleconnections based on sea level pressure are well-known drivers of precipitation anomalies in many world regions^{7,8}. Therefore, we tested for relationships between the reanalysis-referenced F_1 -scores of CME and Taylor S-scores^{9,10} for precipitation rates, which measure grid-cell-wise errors in conjunction with overall discrepancies in precipitation variability across a spatial domain."

We had to rephrase these sentences slightly in response to the second referee.

Re the parabolic relationship: (1) I don't say that we shouldn't aim for this. I just want to make some caveats explicit, and i think you should do that too in the paper that you aim at a rather generic audience. 1. Model projections can be right for the wrong reason (as you wrote yourself). And then it is very unlikely that the projection remains right under new conditions too. 2. Even if the projection is right for the right reason, under

new conditions the model may be wrong. Basically models are not calibrated for future climate scenarios. 3. Perhaps the models that do well in precipitation projections, they do so because they were calibrated with respect to that, or with respect to something closely related, and other models would also do okay had they been calibrated like that. Still, problem 1., 2. remain.

(2) Do you have some explanation why a quadratic relationship should hold? If the relationship is nonlinear, then a quadratic model is just the minimum that you can do. In that case, the expression "there appears to be a parabolic relationship" is not really honest.

Re 'most likely/less likely': I'm pretty sure the idea of probability is different from how you use it here. The probability of tails when flipping a coin is 0.5 if you don't know anything. You can revise this probability based on some predictability, but that predictive information is supposed to be a fact. If you make predictions by a model and have model errors, and you cannot characterise that uncertainty, then you cannot make a statement with respect to probability in a strict sense. Essentially, you are making a statement about your feelings, intuition. It should be straightforward to rephrase here avoiding the use of the words "probable", "likely".

We agree and rephrased the paragraph and also added some additional explanations:

"Finally, we find strong indications that our novel causal metrics could aid in constraining uncertainty in precipitation projections under climate change. As mentioned above, past model skill in a quantity does not automatically imply skill for future projections as models can be right for the wrong reasons. In contrast, the networks we use here infer rather complex dynamical coupling relationships from SLP data that are effectively impossible to calibrate against current observations, different from, for example, quantities such as global surface temperature¹¹. The underlying premise is that today's dependencies between the dynamical mechanisms and e.g. precipitation rates represent actual physical processes that also hold under future climate change. As a result, causal discovery could provide more robust insights by identifying causal mechanisms that are more likely to hold also under future climate change scenarios. It is therefore interesting to consider our complex causal information quantity in terms of constraining future precipitation projections. Indeed, we find no relationship between the past global precipitation skill S-scores and future precipitation rate changes in the CMIP5 projections, but there appears to be an approximately parabolic relationship between projected CMIP5 global land precipitation rate changes attained by the period 2050-2100 (relative to 1860-1910; Supplementary Figure S16) and F₁-scores from historical runs (Figures 4b/c). This implies intermediate model range land precipitation changes of around 0.0-0.1 mm/day according to the causal fingerprint scores, as opposed to the most extreme negative and positive changes. We also note that simpler dynamical metrics, e.g. based on SLP Taylor S-Scores, or the aforementioned storm track skill scores, and using the same non-parametric Gaussian Process regression (Figure 4b/c; Methods), do also not yield such emergent relationships (Figure 4b/c, Supplementary Figures S18-S21).

The possibilities for future projections are constrained by the models participating in CMIP5. Therefore, we can only constrain the relationship within the given data boundaries. Similar model evaluation exercises, also for variables other than precipitation and atmospheric dynamical interactions, could be tested for similar emergent relationships in the ever-expanding data made available through observations and climate modelling projects. Such studies might flexibly combine the blueprint of the method outlined here with other dimension reduction techniques and/or causal discovery algorithms."

p. 10:

Figure 4a: I think the physical quantity and its unit should be indicated here. I see that you are saying it in the caption, but still (a colorbar is no different from the axis of a diagram).

Done.

p. 11:

'teleconnections as a key driver'

We have revised this sentence in response to the other reviewer to: "...suggesting atmospheric dynamical interactions as a key driver of important regional climate model errors".

'We further see great scope in using our framework to better understand differences between models and observations': This has not been demonstrated here, and so i'm not sure why i should believe in this. I mean particularly the "understanding" bit. You applied an algorithm and you haven't interpreted or explained the results much. I'm not saying it's a problem, or your work is not worth publishing, just let's be honest. We know how science is often driven: we have a tool; let's apply it!

Granted. However, as we already pointed out above, we consider our study as a blueprint that outlines a novel method to carry out climate model evaluation. This has already helped us understand that there is a relationship, for example, between how well the dynamical interactions in the atmosphere are modelled and certain regional precipitation errors. The networks constructed could now be compared in detail, which is certainly beyond the scope of this study, but definitely something we want to do in the future. Of course, we also hope to motivate others to look into differences in the networks for models and observations. This is certainly a difficult task, but we also certainly see scope for this, especially given that causal discovery algorithms were designed to identify underlying physical mechanisms (see, e.g., Kretschmer et al. 2016, Runge et al. 2019). Understanding differences in the representation in models and observations is key for model evaluation. This potential is really all that we intend to mention here, as this paragraph is about the broader implications.

p.15:

'physically distinct': I don't understand why a principal component or EOF would carry physical meaning. The EOF is just a mathematical object that breaks down variability in a way that is convenient. When the EOF is applied to a region and a physical quantity (e.g. pressure), it perhaps picks up some physically important pattern. But the pick of the region and quantity/variable is up to the understanding of the physics by the person.

We agree with the general point, but then again - as we also say in the same sentence - we can identify quite clearly certain well-established climate modes such as the ENSO and the NAO in the components. It is not a major point though so we just changed the wording to:

"The dimension reduction step (Figure 1b) serves as a data-driven method to extract large-scale patterns of regional sea-level pressure variability that in many cases resemble well-known climatological processes such as the ENSO or the North Atlantic Oscillation (NAO)."

'we here choose truncated principal component analysis': Why?

No particular reason. If the number of components is chosen by variance or by the number of components doesn't really matter here (it would not change the results or the conclusions much). We just made sure that a very significant portion of the variance in the data was captured by the number of components we defined (see our Supplementary figures). Note also that the variance metric associated with each component becomes meaningless after the Varimax rotation is applied.

'spatially orthogonal': They are orthogonal in a mathematical sense, not in a spatial/geometrical sense.

We have removed the word 'spatially' here.

'noisy components': Should use quotation marks, because you mean noise not in the sense of randomness -- the decomposition can of course be applied to completely deterministic systems. You mean a component of variability whose decorrelation time is much shorter than those of the processes that you are interested in.

Done.

'spatial weights': Why do you use the word "spatial" here? The EOF is a spatial pattern. You just mean the weight or contribution of that pattern to a spatial pattern of interest that the EOF derives from.

We changed this to "spatial pattern (loading)".

Why SLP?

We have added the following explanation:

"The motivation behind using sea level pressure as the variable underlying the networks is that it is a standard variable to characterize large-scale atmospheric dynamics and corresponding variability, e.g. in climate modes or weather patterns. Therefore, it is also available in virtually any reanalysis dataset or model data archive, which allowed us to work with the largest possible number of ensemble members for the CMIP5 analysis."

'The components obtained for the four meteorological seasons for the NCEP data can be found in Supplementary files 2-5': ? ... Okay, now i'm pretty sure you are referring indeed to one of those e.g. 218050_0_supp_3966261_pv7c4r.pdf. Although, it is not possible to verify things, as in these pdf's even the physical quantity is not named (or perhaps the meteorological season is what separates these files). Furthermore, 1. the time series cannot be resolved (at least on my screen) -- "some noisy time series" could economically stand for as many as several hundred diagrams that you supply; at this point it is not clear why you show either 2. the power spectrum or 3. the autocorrelation function. I think instead of these long pdf's, you'd better supply rather the data on one hand and code on the other hand that produced these. Then whoever is interested, would have it and have even more. Okay, i see now that i lost trace of the identification of the Supp' file as file 1, 2, etc. because i downloaded the "Reviewer Zip File".

Apologies for the confusion with the Supplementary files. We will try to resolve this issue during the next upload. The time series themselves of course offer limited insight (apart from the weights magnitude) so that we plotted additionally the power spectrum (important, for example, to identify climate modes such as the ENSO) and the autocorrelation plots (important to consider when identifying causal interactions).

The files themselves are several GB in size and we will certainly make them available with all Jupyter notebooks, Python files and data via the CEDA data archive as soon as this manuscript is published (see our now included data availability statement). However, we would also like to offer the Supplementary pdfs as a visual inspection of how the components appear for those readers who might not be inclined to work on the data or to create figures themselves.

p. 16:

' At this point i don't know what the F-1 score is.'

We have moved the section on the F₁-score up to the top of the Methods section.

'If Y is a scalar, then i suggest you use a lower-case symbol, because upper case traditionally denote vectors or matrices.'

We use the standard notation of statistics here

(https://en.wikipedia.org/wiki/Notation_in_probability_and_statistics): Random variables are usually written in upper case roman letters. Particular realizations of a random variable are written in corresponding lower case letters. Bold face denotes vectors. Here we refer to the random variables X , Y .

'Granger causes': I suppose, without reading your reference (sorry), this expression stands for a specific meaning of causality. I think you should use quotation marks, and perhaps also a hyphen: X "Granger-causes" Y . Btw. i guess this is a quite restrictive sense, as it is based on a specific linear model, the VAR model.

Done.

'if any of the coefficients belonging to X is non-zero': I don't understand what you mean. I do think that what you extract from your reference should be clear without having to consult your reference.

We rephrased this to: "Then X 'Granger-causes' Y if any of the coefficients corresponding to different time lags of X is non-zero (typically tested by an F-test).

'low detection power': I suppose you use some threshold or some significance level to declare a coefficient nonzero -- as you define Granger causality above. That makes sense, as you will never estimate a parameter to

be exactly zero. But at this point i don't know what low detection power means. Also, i suspect that "detection power of Granger causality" is not the most meaningful language, but rather we can speak perhaps about the "detectability of Granger causality".

'if too many variables are conditioned on and for strong autocorrelation': I don't understand this.

We rephrased this to: "As analysed in ref. 21, Granger causality, due to a high model complexity for a given finite sample size, has low detection power for causal links (true positive rate) if too many variables are used and also for strong autocorrelation, both of which are relevant in our analysis."

'The F1-score is a widely used metric for such problems as it balances the statistical precision (P) and recall (R)': Is there any justification for this balance? Or is it just an arbitrary yet not senseless way of coming up with a single instead of two (TP, FN) "objectives". I pointed out in Physica D 313 (2015) 37–50 in the context of "categorical" prediction of threshold exceedances that "It takes a specific application to be possibly able to define a scalar-valued cost function" that is to measure prediction skill. PHYSICAL REVIEW E 96, 032120 (2017) outlines more generically the definition of the prediction skill in terms of the contingency table (Table II) and eqs. (9-14). Of course these are not my original ideas. TP is also called "hit rate";

The use of the F_1 -score is quite common and introduced here as we naturally have a sparse system (few true links in the network). Of course, there will be other scores that might be even better at taking these effects into account for any given problem. We acknowledge this in our revised formulation:

"The F_1 -score is a widely used, however necessarily imperfect⁶⁶, metric for such problems as it balances the statistical precision (P) and recall (R)."

p. 17:

'sign of dependence': This sounds strange. Do you mean perhaps the "directionality of causality" (i've just made the expression up)?

We actually mean the sign (positive or negative) of the (partial) correlation coefficient. This has nothing to do with the direction, which in our framework is assessed based on the time lag. Rather, we want to ensure that a link in network B should not have opposite sign compared to the same link in network A. Only if a model reproduces a teleconnection dependency with the same sign, it is a 'match'.

Is there a weighting here wrt. the area represented by gridpoints?

Thank you for bringing this to our attention. We actually calculated the correlations and standard deviations for the S-score on a grid-by-grid cell basis, whereas everywhere else we took spatial non-uniformity of the grid cells into account. We therefore re-calculated all S-scores and accordingly updated each relevant figure in the main text (Figure 4) and Supplementary (Figures S9-S12). As you can see, there are very small differences in the results and none affect our conclusions. Nevertheless, thank you again as it is certainly a good idea to make this consistent. We have also updated the Methods section with the relevant equations for how we calculated the correlation coefficients and standard deviations with appropriate spatial weighting.

'where R is the pattern correlation coefficient between the models and observations and SDR is the ratio of spatial standard deviations': Similarly as with F1, there needs to be some justification for this particular balancing of two objectives.

We have extended the explanatory sentence following this statement to: "The score thus considers both the pattern similarity over the spatial domain with regard to a given quantity as well as their amplitude ratios, as both the spatial coherence and magnitude range of a variable is important for measuring model skill⁹."

'linearly de-trended': Of course to have true anomalies, you would need to subtract the forced response signal being an ensemble mean. <https://journals.ametsoc.org/doi/full/10.1175/JCLI-D-14-00459.1> However, you don't have it. For the actual Earth system of course we will never have an ensemble, but you don't have large

ensembles even for the CMIP5 models. You say without any justification what you do. I really think you must say why you don't expect much error introduced by this "not ideal" method. It does not appear to me at any rate that it is trivial why. May be my paper on analysing the forced response of teleconnections is of interest to you <https://arxiv.org/abs/1803.08909> We find that in the 20. century the ENSO-Indian monsoon teleconnection changes (at least in a model -- the same question about the truth via observation is open and "challenging"). Therefore, even if the linear detrending wasn't a problem in obtaining genuine anomalies and their fluctuations, treating the 20. c. as if it was stationary (say wrt. the internal variability that the anomalies represent) might be an issue here. For the sake of comparing models it might not be an issue, but ideally the concept with respect to which they are compared is meaningful.

'anomalized': I don't understand this word: does it refer to the way you deal with seasons.

Thank you for this comment as it allows us to clarify a few points.

Firstly, 'anomalized' is indeed a standard term in meteorology to indicate the removal of the seasonal cycle. We are interested in anomalies relative to what would be normal at the given time of the year, which in this case meant that we subtracted the long-term daily mean from all sea level pressure data.

Linear de-trending is certainly imperfect but probably the most prudent choice. We have looked, in a variety of different projects, at very long sea level pressure datasets on grid-cell-wise and larger spatial scales, both for reanalyses and climate model data. In none of these datasets did we find any very large trends in sea level pressure data, simply because this variable is not affected much by the thermodynamic effects of global warming. This even holds true under strongly forced future scenarios, e.g. the RCP8.5 scenario with much larger forcings than what we have observed historically. This is one of the reasons why we choose sea level pressure as a variable in the first place as it primarily reflects dynamical variability. Below we show example surface temperature and sea level pressure time series averaged over European land (courtesy of Carl Thomas, a PhD student at Imperial College) as simulated through a CMIP5 model for the historical forcing scenario (years 1860-2005) and the future large forcing RCP8.5 scenario (from 2005-2100):

Linear fits are applied to different periods of the data, which make clear that for sea level pressure any trends would be much smaller than the internal variability, in contrast to surface temperature. As a result, linear de-trending appears to be a good approximation for sea level pressure to remove any possible trends (e.g. due to possible small model drifts). In reply to these two comments we have revised this sentence to:

"All data (including the reanalysis datasets) was linearly de-trended on a grid cell basis and seasonally anomalized by removing the long-term daily mean. Note that sea level pressure data is effectively stationary even under forced climatic conditions so that the de-trending is a prudent step to remove any potentially occurring small trends to a good approximate degree. Of course, we cannot fully account for the very long time-scales that may be associated with some climate processes⁶⁹ beyond the time-scale covered by each individual dataset."

Reply to Referee 2:

We thank the referee for his/her insightful comments and suggestions, which have helped to further improve our manuscript. In particular, the comments have helped us to add to the physical interpretation of our causal networks. We address each comment (*italic*) in our replies (**bold**) below.

This paper reports findings of an analysis where causal discovery algorithms were applied to identify causal fingerprints in both observations (reanalysis products) and CMIP5 model simulations. The authors conclude that a model's capability of simulating precipitation patterns over selected regions highly correlates with the model's skill of capturing these causal fingerprints. I agree fully with the authors that machine learning has huge potential for contributing to the study of atmospheric processes given the enormous amounts of data available from routine meteorological observations and weather and climate model simulations. However, I have to recommend a rejection of the present manuscript due to the following: "causal teleconnection fingerprints" are stated as the subject of the study yet the authors clearly misunderstood the concept of teleconnection in atmospheric sciences. Teleconnection refers to synchronized changes in monthly (or longer time) mean atmospheric fields (e.g., temperature, pressure or geopotential height) over different geographical locations. Classic examples include the North Atlantic Oscillation (NAO) and Pacific North American pattern (PNA). The 3-day averaged SLP data, after the further dimension reduction via PCA, contains mainly synoptic and sub-monthly-scale variability that are effectively "weather", not classic modes of teleconnection. This explains why the PCA patterns shown in the supplementary plots are all spatially isolated pressure anomalies. Due to the weather, thus short-lived nature of these anomalies, the causal connections among the nodes based upon PCA components are very sparse except for regions where background steering flow is prominent enough to allow upstream weather to cause changes in the weather downstream. This also explains why the time lags identified for these causal connections are mostly 1s (i.e., 3 days) with a few 2s. Since the causal fingerprints detected here are physically weather anomalies that are occasionally connected over a short timescale, there exist much simpler dynamics-based metrics for evaluating model's simulations of these features. For example, one can classify the CMIP5 models simply by calculating the spatial correlations between observed and modeled seasonal mean winds and storm tracks (e.g., in terms of variance of band-passed pressure or height fields). Model's skills of simulating the seasonal mean winds plus storm tracks are highly correlated with model's simulation of precipitation. These metrics are much easier to calculate compared to the causal fingerprints discussed in this paper.

In summary, the authors misinterpreted the identified causal fingerprints as those associated with teleconnections while in fact those are sparse connections between local weather anomalies. Much simpler metrics can be constructed based on the physical nature of these anomalies to classify and evaluate model performances in simulating critical fields such as precipitation. The present manuscript fails to convince this reviewer that we gain anything new by going through this practice. There have also been studies in the past discussing causal connections among atmospheric weather disturbances across daily timescales thus the application itself is not innovative either.

We thank the referee for his/her insightful comments, which raise two main questions:

(1) Is the use of the term 'teleconnections' appropriate here?

(2) As a result, would there be metrics simpler than causal fingerprints which provide the same insights into model performance and projections?

Concerning (1), we have interpreted teleconnections here in the broadest possible sense as large-scale dynamical interactions between remote world regions. For example, the connection between the East Pacific and West Pacific mode of the ENSO in the tropical Pacific would certainly fall under this regime even if the most significant interactions are taking place on weekly timescales. We would also like to highlight that monthly or seasonally averaged anomalies will always include signals due to these short-term interactions so that looking at longer-term average metrics might actually not be entirely different.

The use of causality algorithms here gives us the opportunity to consider not only time-averaged simultaneous correlations, but also time-lagged information. This is an advantage and we want to use this information. It is probably also not surprising that we find that the most significant correlations

typically are found on shorter time-scales, which is entirely intuitive - as is their relevance for precipitation patterns.

To cut a long story short, we certainly include some interactions that could classically be termed teleconnections but also acknowledge that other shorter interactions can be termed as 'weather' as suggested by the referee. We have therefore removed or replaced all uses of the term 'teleconnection(s)' in our manuscript by other phrases such as 'atmospheric dynamical interactions', which also applies to the paper title. In terms of the high localization of the principal components used here, we should add that this behaviour is by design. Localization will primarily be driven by the additional Varimax rotation, which is designed to provide more localized components than standard PCA.

Concerning (2) and novelty aspects, in the revised manuscript we point out more clearly: (a) the novel application of causal discovery algorithms to evaluate climate models (they have been used before but not for this purpose and context), and (b) our demonstration that these algorithms do indeed provide useful metrics which are not only meaningful, as they correlate with the skill of models to reproduce precipitation patterns, but also provide an emergent relationship for future precipitation changes in climate change projections. We are not aware of any simple dynamical metrics that would provide such a relationship for precipitation. We also demonstrate that, for example, past precipitation modelling skill does not straightforwardly transfer into future precipitation projections. As the referee states himself/herself, this is not a trivial statement.

We entirely agree with the referee that storm tracks/waveguides will play a role in the interactions represented in the causal network and it is a very interesting question how they contribute to the correlation with modelled precipitation skill.

Motivated by the referee's comments, we tested several simple metrics with regard to their correlation with precipitation and their implications for future precipitation rate changes predicted by each model. For this, we essentially reproduce Figure 4 in the main text with each of those metrics taking the place of the F_1 -score one at a time. The overarching conclusions are that we find lower correlations with the historical precipitation modelling scores and also, in all cases, no apparent relationship with future precipitation changes. This highlights the advantages and underlines the meaningfulness of our novel causal model evaluation approach. Having said that, in particular a classic storm track metric showed some significant correlations with precipitation skill (as suggested by the referee) and this analogy has motivated us to further add to the scientific interpretation of our causal networks, see below.

In terms of the metrics we considered, these are:

1) A Taylor S-score purely based on sea level pressure. We first calculated such scores for annual mean sea level pressure patterns. As we show in Supplementary Figure S18, this provides much worse correlations with Taylor precipitation skill scores, which are typically non-significant at standard significance levels and we also find no clear emergent relationship with future precipitation rates changes.

2) As 1) but for seasonal averages, i.e.:

$$S(\text{slp, annual}) = (S(\text{slp, DJF}) + S(\text{slp, MAM}) + S(\text{slp, JJA}) + S(\text{slp, SON}))/4$$

Again, we found low correlations with precipitation skill scores and no emergent relationship with regard to future precipitation changes (Supplementary Figure S19).

3) A classic storm track skill score which is based on pattern correlations between observed and modelled grid-cell-wise 2-6-day bandpass-filtered sea level pressure standard deviations (Ulbrich et al., 2008).

Ulbrich et al. Changing Northern Hemisphere storm tracks in an ensemble of IPCC Climate Change simulations. *Journal of Climate* 21, 1669-1679 (2008).

This metric indeed provided much better correlations with past precipitation skill scores than methods (1) and (2), which underlines the referee's point that storm tracks are an important physical feature for our causal networks (Supplementary Figure S20). We found this aspect interesting, because our initial submission did not discuss these physical aspects sufficiently. At the same time, storm tracks are not the only features picked up by our causal discovery method as is evident from the facts that (a) the causal network skill scores still show higher correlations with the precipitation skill scores and (b) the storm track skill scores do not provide the same emergent relationship for future changes. However, as would be expected from the prominent role storm tracks appear to play in our networks, we do find that the results concerning future changes also come closer to the ones obtained with our causal fingerprint scores. We therefore added a sentence to the caption of Figure 4 to reflect these important conclusions and also added the following paragraph on the interpretation of the causal networks to our manuscript:

"An interesting question is how to interpret the relationship between precipitation and the causal network skill scores from a physical point of view. Notably, the causal networks are, especially at stringent significance thresholds, dominated by interactions on a timescale of less than one week (lag $\tau \leq 2$; Figure 1d). This timescale is broadly equivalent to dynamical interactions related to storm tracks⁵⁷. Simple metrics have been used before to quantify the skill of climate models to capture storm tracks, e.g. pattern correlations in standard deviations of 2-6-days bandpass-filtered daily mean SLP data⁵⁸. Indeed, Taylor S-scores for precipitation are also positively correlated with such simpler metrics (Supplementary Figures S18-S20), which altogether indicates that a large part of the links in the causal networks represent dynamical interactions related to storm tracks. At the same time, these simpler metrics generally show smaller and less significant correlations with the precipitation scores on a global as well as on regional scales, highlighting that our causal networks identify additional relationships which further improve the correlations with precipitation. Longer time-scale dynamical interactions, for example triggered by the ENSO and its zonal couplings, as well as effects on the extratropics, are prime candidates for explaining some of the higher skill related to our causal fingerprint scores."

We further note that our complex metric also has several other preferable features. For instance, it is practically impossible to calibrate for, unlike many simpler metrics used in climate science nowadays, a point we partly make in response to Referee #1. Finally, we also added a comment on the potential of our metric to constrain future projections as compared to the simple storm track metric:

"Finally, we find strong indications that our novel causal metrics could aid in constraining uncertainty in precipitation projections under climate change. As mentioned above, past model skill in a quantity does not automatically imply skill for future projections as models can be right for the wrong reasons. In contrast, the networks we use here infer rather complex dynamical coupling relationships from SLP data that are effectively impossible to calibrate against current observations, different from, for example, quantities such as global surface temperature¹¹. The underlying premise is that today's dependencies between the dynamical mechanisms and e.g. precipitation rates represent actual physical processes that also hold under future climate change. As a result, causal discovery could provide more robust insights by identifying causal mechanisms that are more likely to hold also under future climate change scenarios. It is therefore interesting to consider our complex causal information quantity in terms of constraining future precipitation projections. Indeed, we find no relationship between the past global precipitation skill S-scores and future precipitation rate changes in the CMIP5 projections, but there appears to be an approximately parabolic relationship between projected CMIP5 global land precipitation rate changes attained by the period 2050-2100 (relative to 1860-1910; Supplementary Figure S16) and F_1 -scores from historical runs (Figures 4b/c). This implies intermediate model range land precipitation changes of around 0.0-0.1 mm/day according to the causal fingerprint scores, as opposed to the most extreme negative and positive changes. We also note that simpler dynamical metrics, e.g. based on SLP Taylor S-Scores, or the aforementioned storm track skill scores, and using the same non-parametric Gaussian Process regression (Figure 4b/c; Methods), do also not yield such emergent relationships (Figure 4b/c, Supplementary Figures S18-S21)."

We understand that the referee took a rather critical perspective on our original manuscript and we have found this perspective very helpful, leading to our additional analysis which has helped us to

improve the physical interpretability of our networks. Our causal discovery metric is still superior to the simple metrics we added in terms of correlations with past precipitation scores and future projected changes. Therefore, we can draw only positive conclusions and hope that the referee will now share our view on the usefulness of our results.

REVIEWERS' COMMENTS:

Reviewer #1 (Remarks to the Author):

This is my second review of the manuscript. Overall I'm satisfied with the authors' responses and considerable extra work on the manuscript. 1. It is reassuring that the weighting of spatial correlations with respect to the grid cell areas (Eq. (6)) did not alter the results notably; and 2. that the linear detrending of sea level pressure time series appears to be a reasonable choice. Furthermore, 3. they demonstrate the somewhat better performance of the "causal dynamical characteristics" ("fingerprint") compared to simpler statistics, e.g. to do with the representation of storm tracks, suggested by the other referee, in prompting a better modelling of precipitation. I'm also delighted to see that this has led to some, even if limited, physical insight.

Regarding my personal scientific interest, I'm intrigued by the possibility of the straightforward application of their method to what the other referee would rather call teleconnections — relationships on larger spatial and longer time scales.

Otherwise, I have come to appreciate a bit more, I think, a caveat about the author's suggestion about constraining precipitation change projections into the future. In my previous review regarding the parabolic relationship seen in Fig. 4b I wrote the following:

"I don't say that we shouldn't aim for this. I just want to make some caveats explicit, and I think you should do that too in the paper that you aim at a rather generic audience. 1. Model projections can be right for the wrong reason (as you wrote yourself). And then it is very unlikely that the projection remains right under new conditions too. 2. Even if the projection is right for the right reason, under new conditions the model may be wrong. Basically models are not calibrated for future climate scenarios. 3. Perhaps the models that do well in precipitation projections, they do so because they were calibrated with respect to that, or with respect to something closely related, and other models would also do okay had they been calibrated like that. Still, problem 1., 2. remain."

It is point 3. that is eliminated given that the "causal dynamical characteristics" (probably) cannot be calibrated for. But I did write explicitly that problem 1. 2. remain. I suppose point 1. is indeed also eliminated, as I suppose that a model that gets the "causal dynamical characteristics" right does get the precipitation right for the right reason. Now, what's the situation with point 2.?

The modelled precipitation is an emergent characteristics just like the "causal dynamical characteristics". Both of these emergent characteristics seem to depend on parametrizations, or "model developmental choices". The property of causality does not imply physical realism. In a way, it is a mathematical property. Phenomenological models — which can be quite wrong — will feature their own "causal dynamical characteristics". Therefore, the caveat of point 2. seem to remain:

models that are describing the historical climate reasonably well for the right reason, including the representation of “causal dynamical characteristics”, might not describe well the climate into the future. I see no reason why they should be more robust under climate change than other emergent characteristics. (Whether these characteristics can or cannot be calibrated against seems to be irrelevant to this point.) This robustness is not argued for by the authors; on line 223 they just say it is a “premise”, but not why this premise is justified. Am I missing something?

If this is right, then there is no reason to think that the model with the largest F1 score based on historical data will best predict/project the future precipitation change — as an intermediate value/range if the F1-dependence on the precipitation change is a downward parabola. The idea of constraining (based on the largest F1 values) itself is not wrong, just that we cannot be sure what is the correct relationship to use for constraining (i.e. how is the parabola in Fig 4b might be distorted under future conditions).

The authors should either argue more convincingly about this /“explain it better to a nonspecialist”, or do state this caveat, as I suggested to them.

Tamas Bodai

Note: My policy is to not make a recommendation to editors on the publication of manuscripts. It is the editor's duty to make up their mind based on (ideally factual) referee reports, or one that reflects the referee's (ideally unbiased) opinion.

Reviewer #2 (Remarks to the Author):

I appreciate the authors' effort to clarify the terminology of “teleconnection”, rephrase relevant statements and also include additional analyses of model evaluation metrics such as storm tracks. The manuscript has improved quite a bit with these revisions. However, I could not recommend acceptance of the manuscript in its present form due to the authors' completely ignoring two important papers published more than five years ago actually for the first time reporting evaluations of climate models in a causality-based framework. Specifically, Ebert-Uphoff and Deng (2012) and Deng and Ebert-Uphoff (2014) for the first time introduced a causal-discovery based climate network, defined terms such as “local memory” and “remote impact” of network nodes (the present manuscript has similar terms, “self-links” and “cross-MCI”). These two earlier papers also used “information pathway” to summarize the general characteristics of the network while the present manuscript invoked the terminology of “causal fingerprints”. Deng and Ebert-Uphoff (2014) further demonstrated that one of the CMIP5 models (NCAR CCSM4) was capable of reproducing general features of the observed information pathway and illustrated key changes of such information pathway as the climate warms. Therefore the efforts of Deng and Ebert-Uphoff (2014) also constitute a real first step in systematic model evaluation in a causality-based framework.

Furthermore, synoptic-scale disturbance as the key information-carrier in the system is one of the main discoveries reported by Ebert-Uphoff and Deng (2012) and Deng and Ebert-Uphoff (2014), which was basically confirmed by findings in the present manuscript where much of the causal footprints identified were shown to be related to weather anomalies and storm tracks.

Given all of these, it is shocking to see the authors not citing these earlier work and make the following claim “We apply for the first time data-driven causal discovery algorithms to identify unique causal fingerprints characterizing atmospheric dynamical interactions”. In my view, a thorough discussion and a complete and accurate acknowledgement of these earlier work must be done before the manuscript could be accepted by any reputable journals. The authors also need to clearly emphasize conceptual and/or methodological differences (if any) with the earlier work and discuss real advances being made in the present study.

There are some other caveats in the analysis approach adopted here. For example, carrying out the analysis on a global domain actually led to harder physical interpretations of the findings since atmospheric dynamical interactions leading to precipitation are distinctly different in scales between winter and summer, and DJF (JJA) corresponds to the Northern Hemisphere winter (summer) and Southern Hemisphere summer (winter). Caveats like these are partly responsible for the trivial nature of some of the results (especially those provided in the supplemental materials), which was also pointed out the other reviewer.

References:

- 1.Ebert-Uphoff, I. and Y. Deng, 2012: A new type of climate network based on probabilistic graphical models: Results of boreal winter versus summer, *Geophysical Research Letters*, 39, L19701, doi:10.1029/2012GL053269.
- 2.Deng, Y., and I. Ebert-Uphoff, 2014: Weakening of atmospheric information flow in a warming climate in the Community Climate System Model. *Geophysical Research Letters*. 41, 193–200, doi:10.1002/2013GL058646.

31 January 2020

Reply to the Referees - Nature Communications Manuscript NCOMMS-19-25061-A

Below we reply to each of the two referee's comments (*italic*) point-by-point in **bold** font.

Reply to Referee #1:

We thank Dr. Bodai for his second round of constructive and overall positive review comments. We hope that he will find our reply to his last minor comment satisfactory.

This is my second review of the manuscript. Overall I'm satisfied with the authors' responses and considerable extra work on the manuscript. 1. It is reassuring that the weighting of spatial correlations with respect to the grid cell areas (Eq. (6)) did not alter the results notably; and 2. That the linear detrending of sea level pressure time series appears to be a reasonable choice. Furthermore, 3. they demonstrate the somewhat better performance of the "causal dynamical characteristics" ("fingerprint") compared to simpler statistics, e.g. to do with the representation of storm tracks, suggested by the other referee, in prompting a better modelling of precipitation. I'm also delighted to see that this has led to some, even if limited, physical insight.

Regarding my personal scientific interest, I'm intrigued by the possibility of the straightforward application of their method to what the other referee would rather call teleconnections - relationships on larger spatial and longer time scales.

Otherwise, I have come to appreciate a bit more, I think, a caveat about the author's suggestion about constraining precipitation change projections into the future. In my previous review regarding the parabolic relationship seen in Fig. 4b I wrote the following:

"I don't say that we shouldn't aim for this. I just want to make some caveats explicit, and i think you should do that too in the paper that you aim at a rather generic audience. 1. Model projections can be right for the wrong reason (as you wrote yourself). And then it is very unlikely that the projection remains right under new conditions too. 2. Even if the projection is right for the right reason, under new conditions the model may be wrong. Basically models are not calibrated for future climate scenarios. 3. Perhaps the models that do well in precipitation projections, they do so because they were calibrated with respect to that, or with respect to something closely related, and other models would also do okay had they been calibrated like that. Still, problem 1., 2. remain."

It is point 3. that is eliminated given that the "causal dynamical characteristics" (probably) cannot be calibrated for. But I did write explicitly that problem 1. 2. remain. I suppose point 1. is indeed also eliminated, as I suppose that a model that gets the "causal dynamical characteristics" right does get the precipitation right for the right reason. Now, what's the situation with point 2.?

The modelled precipitation is an emergent characteristics just like the "causal dynamical characteristics". Both of these emergent characteristics seem to depend on parametrizations, or "model developmental choices". The property of causality does not imply physical realism. In a way, it is a mathematical property. Phenomenological models — which can be quite wrong — will feature their own "causal dynamical characteristics". Therefore, the caveat of point 2. seem to remain: models that are describing the historical climate reasonably well for the right reason, including the representation of "causal dynamical characteristics", might not describe well the climate into the future. I see no reason why they should be more robust under climate change than other emergent characteristics. (Whether these characteristics can or cannot be calibrated against seems to be irrelevant to this

point.) This robustness is not argued for by the authors; on line 223 they just say it is a “premise”, but not why this premise is justified. Am I missing something?

If this is right, then there is no reason to think that the model with the largest F1 score based on historical data will best predict/project the future precipitation change — as an intermediate value/range if the F1-dependence on the precipitation change is a downward parabola. The idea of constraining (based on the largest F1 values) itself is not wrong, just that we cannot be sure what is the correct relationship to use for constraining (i.e. how is the parabola in Fig 4b might be distorted under future conditions).

The authors should either argue more convincingly about this /“explain it better to a nonspecialist”, or do state this caveat, as I suggested to them.

Thank you, this is a valid concern, which we certainly want to address in our revised manuscript.

On the one hand, it seems intuitive that dynamical processes which are important for present-day precipitation modelling are also important for future changes in precipitation. Our causal networks are designed to infer dynamical coupling relationships driven by these processes and pose particularly strong demands on the quality of the model data (e.g. stronger than the simple correlation metrics we also investigate in our manuscript, which again could be more easily right for the wrong reasons). To a degree, the apparent structure we find in Figure 4b appears to validate this assumption, because if there was no relationship between past skill as measured through the F₁-score and future precipitation changes, we would not expect to see a parabolic relationship. Instead, we would rather expect to see a random relationship as for the simple metric found in Figure 4c.

On the other hand, it is also clear that our method/assumption/premise cannot handle processes that are generally not (well) represented in climate models as off today, but which might become important for future changes in land surface precipitation. Similarly, there might be processes that are not dynamical, which might become increasingly important for land precipitation in the future. These obvious potential ‘blind spots’ in our methodology clearly deserve to be mentioned for the general reader. We have therefore, firstly, added a sentence to the paragraph on the discussion of metrics:

“As mentioned above, past model skill in a quantity does not automatically imply skill for future projections as models can be right for the wrong reasons. The networks we use here infer rather complex dynamical coupling relationships from sea level pressure data that are effectively impossible to calibrate against current observations, different from, for example, quantities such as global surface temperature¹¹. Causal discovery methods could thus provide more robust insights by identifying dynamical coupling mechanisms arising from underlying physical processes that are more likely to hold also under future climate change scenarios (see Discussion).”

Further below we have, secondly, extended the paragraph on caveats with regards to future projections:

“Any method resting on the assumption that past model skill in a certain metric can be related to projected future changes necessarily suffers from certain restrictions. Firstly, there could be processes that are not at all (or not well) represented in climate models today, which might become important in the future. However, this is true for any emergent relationship based on model evaluation against past observations. Secondly, not all relevant processes might be well-captured through the chosen metric. Our metric here is focused on dynamical processes (although it might, at least indirectly, capture the effects of some thermodynamical processes^{14,60}), whereas, for example, future changes in soil moisture are probably primarily thermodynamically driven. Future changes in soil moisture, in turn, could regionally modulate future changes in land precipitation⁶¹. Finally, the possibilities for future projections are also constrained by the models participating in CMIP5. Therefore, we can only constrain the relationship within the given data boundaries, and it should be further verified across other scenarios and ensembles (such as CMIP6). Similar model evaluation exercises, also concerning variables other than precipitation and atmospheric dynamical interactions, could test for similar emergent relationships in the ever-expanding data made available through observations and climate modelling projects. Such studies might flexibly combine the

blueprint of the method outlined here with other dimension reduction techniques and/or causal discovery algorithms^{32,33}. “

Finally, we have added the following paragraph to the Discussion section:

“More realistic fingerprints appear to also have implications for projected future changes in land surface precipitation. Causal network analyses could therefore be a promising tool to constrain climate change projections. The underlying premise is that physical processes (e.g., convection, cloud formation, the large-scale circulation) lead to dynamical coupling mechanisms in Earth’s atmosphere. CME aims at statistically representing these couplings in the form causal networks, which in turn are, as we show here, indicative of modelling skill in precipitation. It appears intuitive that modelling skill as captured through our causal fingerprint scores is therefore also relevant for modelling future changes in precipitation, at least so far as the physical processes relevant for present-day precipitation remain important in future climates.”

Reply to Referee #2:

We thank the referee for the overall positive comments and additional suggestions that helped us to put our results into a wider context.

I appreciate the authors' effort to clarify the terminology of "teleconnection", rephrase relevant statements and also include additional analyses of model evaluation metrics such as storm tracks. The manuscript has improved quite a bit with these revisions. However, I could not recommend acceptance of the manuscript in its present form due to the authors' completely ignoring two important papers published more than five years ago actually for the first time reporting evaluations of climate models in a causality-based framework. Specifically, Ebert-Uphoff and Deng (2012) and Deng and Ebert-Uphoff (2014) for the first time introduced a causal-discovery based climate network, defined terms such as "local memory" and "remote impact" of network nodes (the present manuscript has similar terms, "self-links" and "cross-MCI"). These two earlier papers also used "information pathway" to summarize the general characteristics of the network while the present manuscript invoked the terminology of "causal fingerprints". Deng and Ebert-Uphoff (2014) further demonstrated that one of the CMIP5 models (NCAR CCSM4) was capable of reproducing general features of the observed information pathway and illustrated key changes of such information pathway as the climate warms. Therefore the efforts of Deng and Ebert-Uphoff (2014) also constitute a real first step in systematic model evaluation in a causality-based framework. Furthermore, synoptic-scale disturbance as the key information-carrier in the system is one of the main discoveries reported by Ebert-Uphoff and Deng (2012) and Deng and Ebert-Uphoff (2014), which was basically confirmed by findings in the present manuscript where much of the causal footprints identified were shown to be related to weather anomalies and storm tracks.

Given all of these, it is shocking to see the authors not citing these earlier work and make the following claim "We apply for the first time data-driven causal discovery algorithms to identify unique causal fingerprints characterizing atmospheric dynamical interactions". In my view, a thorough discussion and a complete and accurate acknowledgement of these earlier work must be done before the manuscript could be accepted by any reputable journals. The authors also need to clearly emphasize conceptual and/or methodological differences (if any) with the earlier work and discuss real advances being made in the present study.

There are some other caveats in the analysis approach adopted here. For example, carrying out the analysis on a global domain actually led to harder physical interpretations of the findings since atmospheric dynamical interactions leading to precipitation are distinctly different in scales between winter and summer, and DJF (JJA) corresponds to the Northern Hemisphere winter (summer) and Southern Hemisphere summer (winter). Caveats like these are partly responsible for the trivial nature of some of the results (especially those provided in the supplemental materials), which was also pointed out the other reviewer.

Thank you for pointing us towards these two important references, which we now cite at several places throughout the manuscript. We have also reworded the abstract, partly in response to some editorial comments and suggestions. In addition, we discuss differences between the network methods in the Method section and point towards this discussion within the main manuscript where we discuss the role of storm tracks. One difference between our study and these two previous papers is that we learn networks from two reanalysis datasets and compare them to a large set of climate model simulations, whereas the two previous studies only considered interactions within observations (which we have also done before), or evaluated modelled interactions against observations only for one individual model. From the ensemble of climate models, we can find the emergent relationships with future precipitation changes. Such emergent constraints arising from network studies have not been discussed before. We also use different metrics/methods to assess the model-model and model-observation differences.

Specifically, we have implemented/revised the following paragraphs:

- 1) We cite the two references the first time when discussing the advantages of causal discovery methods in the introduction.**
- 2) We set our storm track discussion into direct context through the following new paragraph:**

"This result is in agreement with earlier work by Ebert-Uphoff and Deng^{32,33} who constructed networks from DJF and JJA NCEP-NCAR reanalysis geopotential height data, as well as from

equivalent data from a single climate model. In their network analyses, they also found storm tracks to be a key driver of network connectivity (see Methods for a comparison of our network methodologies).”

- 3) In the Discussion section, we now highlight these two studies as two central pieces of previous work on causal model evaluation:

“Our work builds on several previous causal network studies in climate science, which were typically focused on network algorithm applications to individual climate modelling or reanalysis datasets, or on the evaluation of dynamical interactions within individual climate models (e.g. refs. 27,32,33,63).

- 4) We have added a full section discussing differences in network methodologies in the Methods section:

“Other network construction methods. As discussed in the main text, causal networks have been used several times before in the climate context. Two of the most prominent cases of such studies are those described in refs. 32,33, where Ebert-Uphoff and Deng also discuss remote impacts and information pathways as well as the role of storm tracks as important drivers of network connectivity. Their work is further a good demonstration of other possible ways to construct causal networks, the effect of which might be an interesting topic for future studies. For example, their network approach was carried out on a grid-cell-wise level rather than using PCA Varimax components. The latter are designed to capture distinct regional climatological processes while an analysis at the grid-cell level is more granular which, however, carries the challenges of higher dimensionality, will have a strong redundancy among neighbouring grid cells, and grid-level metrics will require handling varying spatial resolution among datasets. Furthermore, the original PC causal discovery algorithm used in their work is less suited for the time series case than PCMCI²³. They also used another meteorological variable (500 hPa geopotential height) to construct their networks and compared aggregate network metrics rather than comparing networks on a link-by-link basis.”